# ALINE: Joint Amortization for Bayesian Inference and Active Data Acquisition

**Daolang Huang**[1,2]   **Xinyi Wen**[1,2,3]   **Ayush Bharti**[2]   **Samuel Kaski**[*1,2,4]   **Luigi Acerbi**[*3]

[1] ELLIS Institute Finland

[2] Department of Computer Science, Aalto University, Finland

[3] Department of Computer Science, University of Helsinki, Finland

[4] Department of Computer Science, University of Manchester, UK

`{daolang.huang, xinyi.wen, ayush.bharti, samuel.kaski}@aalto.fi`
`luigi.acerbi@helsinki.fi`

## Abstract

Many critical applications, from autonomous scientific discovery to personalized medicine, demand systems that can both strategically acquire the most informative data and instantaneously perform inference based upon it. While amortized methods for Bayesian inference and experimental design offer part of the solution, neither approach is optimal in the most general and challenging task, where new data needs to be collected for instant inference. To tackle this issue, we introduce the Amortized Active Learning and Inference Engine (ALINE), a unified framework for amortized Bayesian inference and active data acquisition. ALINE leverages a transformer architecture trained via reinforcement learning with a reward based on *self-estimated* information gain provided by its own integrated inference component. This allows it to strategically query informative data points while simultaneously refining its predictions. Moreover, ALINE can *selectively* direct its querying strategy towards specific subsets of model parameters or designated predictive tasks, optimizing for posterior estimation, data prediction, or a mixture thereof. Empirical results on regression-based active learning, classical Bayesian experimental design benchmarks, and a psychometric model with selectively targeted parameters demonstrate that ALINE delivers both instant and accurate inference along with efficient selection of informative points.

## 1 Introduction

Bayesian inference [28] and Bayesian experimental design [61] offer principled mathematical means for reasoning under uncertainty and for strategically gathering data, respectively. While both are foundational, they introduce notorious computational challenges. For example, in scenarios with continuous data streams, repeatedly applying gold-standard inference methods such as Markov Chain Monte Carlo (MCMC) [13] to update posterior distributions can be computationally demanding, leading to various approximate sequential inference techniques [10, 18], yet challenges in achieving both speed and accuracy persist. Similarly, in Bayesian experimental design (BED) or Bayesian active learning (BAL), the iterative estimation and optimization of design objectives can become costly, especially in sequential learning tasks requiring rapid design decisions [21, 30, 55], such as in psychophysical experiments where the goal is to quickly infer the subject's perceptual or cognitive parameters [70, 57]. Moreover, a common approach in BED involves greedily optimizing for single-

---

*Equal contribution.

39th Conference on Neural Information Processing Systems (NeurIPS 2025).

Table 1: Comparison of different methods for amortized Bayesian inference and data acquisition.

| Method | Amortized Inference | | Amortized Data Acquisition | | |
| --- | --- | --- | --- | --- | --- |
| | Posterior | Predictive | Posterior | Predictive | Flexible |
| Neural Processes [27, 26, 52, 53] | ✗ | ✓ | ✗ | ✗ | ✗ |
| Neural Posterior Estimation [49, 54, 34, 60] | ✓ | ✗ | ✗ | ✗ | ✗ |
| DAD [23], RL-BOED [9] | ✗ | ✗ | ✓ | ✗ | ✗ |
| RL-sCEE [8], vsOED [65] | ✓ | ✗ | ✓ | ✗ | ✗ |
| AAL [46] | ✗ | ✗ | ✗ | ✓ | ✗ |
| JANA [59], Simformer [31], ACE [15] | ✓ | ✓ | ✗ | ✗ | ✗ |
| ALINE (this work) | ✓ | ✓ | ✓ | ✓ | ✓ |

step objectives, such as the Expected Information Gain (EIG), which measures the anticipated reduction in uncertainty. However, this leads to *myopic* designs that may be suboptimal, which do not fully consider how current choices influence future learning opportunities [23].

To address these issues, a promising avenue has been the development of *amortized* methods, leveraging recent advances in deep learning [75]. The core principle of amortization involves pre-training a neural network on a wide range of simulated or existing problem instances, allowing the network to "meta-learn" a direct mapping from a problem's features (like observed data) to its solution (such as a posterior distribution or an optimal experimental design). Consequently, at deployment, this trained network can provide solutions for new, related tasks with remarkable efficiency, often in a single forward pass. Amortized Bayesian inference (ABI) methods—such as neural posterior estimation [49, 54, 34, 60] that target the posterior, or neural processes [27, 26, 52, 53] that target the posterior predictive distribution—yield almost instantaneous results for new data, bypassing MCMC or other approximate inference methods. Similarly, methods for amortized data acquisition, including amortized BED [23, 41, 40, 9] and BAL [46], instantaneously propose the next design that targets the learning of the posterior or the posterior predictive distribution, respectively, using a deep policy network—bypassing the iterative optimization of complex information-theoretic objectives.

While amortized approaches have significantly advanced Bayesian inference and data acquisition, progress has largely occurred in parallel. ABI offers rapid inference but typically assumes passive data collection, not addressing strategic data acquisition under limited budgets and data constraints common in fields such as clinical trials [7] and material sciences [48, 44]. Conversely, amortized data acquisition excels at selecting informative data points, but often the subsequent inference update based on new data is not part of the amortization, potentially requiring separate, costly procedures like MCMC. This separation means the cycle of efficiently deciding what data to gather and then instantaneously updating beliefs has not yet been seamlessly integrated. Furthermore, existing amortized data acquisition methods often optimize for information gain across *all* model parameters or a fixed predictive target, lacking the flexibility to selectively target specific subsets of parameters or adapt to varying inference goals. This is a significant drawback in scenarios with nuisance parameters [58] or when the primary interest lies in particular aspects of the model or predictions—which might not be fully known in advance. A unified framework that jointly amortizes both active data acquisition and inference, while also offering flexible acquisition goals, would therefore be highly beneficial.

In this paper, we introduce **A**mortized Active **L**earning and **IN**ference **E**ngine (**ALINE**), a novel framework designed to overcome these limitations by unifying amortized Bayesian inference and active data acquisition within a single, cohesive system (Table 1). ALINE utilizes a transformer-based architecture [69] that, in a single forward pass, concurrently performs posterior estimation, generates posterior predictive distributions, and decides which data point to query next. Critically, and in contrast to existing methods, ALINE offers *flexible, targeted acquisition*: it can dynamically adjust at runtime its data-gathering strategy to focus on any specified combination of model parameters or predictive tasks. This is enabled by an attention mechanism allowing the policy to condition on specific inference goals, making it particularly effective in the presence of nuisance variables [58] or for focused investigations. ALINE is trained using a self-guided reinforcement learning objective; the reward is the improvement in the log-probability of its own approximate posterior over the selected targets, a principle derived from variational bounds on the expected information gain [24]. Extensive experiments on diverse tasks demonstrate ALINE's ability to simultaneously deliver fast, accurate inference and rapidly propose informative data points.

## 2 Background

Consider a parametric conditional model defined on some space $\mathcal{Y} \subseteq \mathbb{R}^{d_{\mathcal{Y}}}$ of output variables $y$ given inputs (or covariates) $x \in \mathcal{X} \subseteq \mathbb{R}^{d_{\mathcal{X}}}$, and parameterized by $\theta \in \Theta \subseteq \mathbb{R}^L$. Let $\mathcal{D}_T = \{(x_i, y_i)\}_{i=1}^T$ be a collection of $T$ data points (or *context*) and $p(\mathcal{D}_T \mid \theta) \equiv p(y_{1:T} \mid x_{1:T}, \theta)$ denote the likelihood function associated with the model, which we assume to be well-specified in this paper (i.e., it matches the true data generation process). Given a prior distribution $p(\theta)$, the classical Bayesian inference or prediction problem involves estimating either the *posterior* distribution $p(\theta \mid \mathcal{D}_T) \propto p(\mathcal{D}_T \mid \theta)p(\theta)$, or the *posterior predictive* distribution $p(y_{1:M}^\star \mid x_{1:M}^\star, \mathcal{D}_T) = \mathbb{E}_{p(\theta \mid \mathcal{D}_T)}[p(y_{1:M}^\star \mid x_{1:M}^\star, \theta, \mathcal{D}_T)]$ over *target outputs* $y_{1:M}^\star := (y_1^\star, ..., y_M^\star)$ corresponding to a given set of *target inputs* $x_{1:M}^\star := (x_1^\star, ..., x_M^\star)$. Estimating these quantities repeatedly via approximate inference methods such as MCMC can be computationally costly [28], motivating the need for amortized inference methods.

**Amortized Bayesian inference (ABI).** ABI methods involve training a conditional density network $q_\phi$, parameterized by learnable weights $\phi$, to approximate either the posterior predictive distribution $q_\phi(y_{1:M}^\star | x_{1:M}^\star, \mathcal{D}_T) \approx p(y_{1:M}^\star | x_{1:M}^\star, \mathcal{D}_T)$ [26, 42, 33, 11, 38, 12, 53, 52], the joint posterior $q_\phi(\theta \mid \mathcal{D}_T) \approx p(\theta \mid \mathcal{D}_T)$ [49, 54, 34, 60], or both [59, 31, 15]. These networks are usually trained by minimizing the negative log-likelihood (NLL) objective with respect to $\phi$:

$$\mathcal{L}(\phi) = \begin{cases} -\mathbb{E}_{p(\theta)p(\mathcal{D}_T \mid \theta)p(x_{1:M}^\star, y_{1:M}^\star \mid \theta)}\left[\log q_\phi(y_{1:M}^\star \mid x_{1:M}^\star, \mathcal{D}_T)\right], & \text{(predictive tasks)} \\ -\mathbb{E}_{p(\theta)p(\mathcal{D}_T \mid \theta)}\left[\log q_\phi(\theta \mid \mathcal{D}_T)\right], & \text{(posterior estimation)} \end{cases} \quad (1)$$

where the expectation is over datasets simulated from the generative process $p(\mathcal{D}_T \mid \theta)p(\theta)$. Once trained, $q_\phi$ can then perform instantaneous approximate inference on new contexts and unseen data points with a single forward pass. However, these ABI methods do not have the ability to strategically collect the most informative data points to be included in $\mathcal{D}_T$ in order to improve inference outcomes.

**Amortized data acquisition.** BED [47, 14, 63, 61] methods aim to sequentially select the next input (or design parameter) $x$ to query in order to maximize the *Expected Information Gain* (EIG), that is, the information gained about parameters $\theta$ upon observing $y$:

$$\text{EIG}(x) := \mathbb{E}_{p(y \mid x)}\left[H[p(\theta)] - H[p(\theta \mid x, y)]\right], \quad (2)$$

where $H$ is the Shannon entropy $H[p(\cdot)] = -\mathbb{E}_{p(\cdot)}[\log p(\cdot)]$. Directly computing and optimizing EIG sequentially at each step of an experiment is computationally expensive due to the nested expectations, and leads to myopic designs. Amortized BED methods address these limitations by offline learning a design policy network $\pi_\psi : \mathcal{X} \times \mathcal{Y} \to \mathcal{X}$, parameterized by $\psi$ [23, 40, 9], such that at any step $t$ the policy $\pi_\psi$ proposes a query $x_t \sim \pi_\psi(\cdot \mid \mathcal{D}_{t-1}, \theta)$ to acquire a data point $y_t \sim p(y \mid x_t, \theta)$, forming $\mathcal{D}_t = \mathcal{D}_{t-1} \cup \{(x_t, y_t)\}$. To propose non-myopic designs, $\pi_\psi$ is trained by maximizing tractable lower bounds of the total EIG over $T$-step sequential trajectories generated by the policy $\pi_\psi$:

$$\text{sEIG}(\psi) = \mathbb{E}_{p(\mathcal{D}_T \mid \pi_\psi)}\left[H[p(\theta)] - H[p(\theta \mid \mathcal{D}_T)]\right]. \quad (3)$$

By pre-compiling the design strategy into the policy network, amortized BED methods allow for near-instantaneous design proposals during the deployment phase via a fast forward pass. Typically, these amortized BED methods are designed to maximize information gain about the full set of model parameters $\theta$. Separately, for applications where the primary interest lies in reducing predictive uncertainty rather than parameter uncertainty, objectives like the *Expected Predictive Information Gain* (EPIG) [67] have been proposed, so far in non-amortized settings:

$$\text{EPIG}(x) = \mathbb{E}_{p_\star(x^\star)p(y \mid x)}\left[H[p(y^\star \mid x^\star)] - H[p(y^\star \mid x^\star, x, y)]\right]. \quad (4)$$

This measures the EIG about predictions $y^\star$ at target inputs $x^\star$ drawn from a target input distribution $p_\star(x^\star)$. Notably, current amortized data acquisition methods are inflexible: they are generally trained to learn about all parameters $\theta$ (via objectives like sEIG) and lack the capability to dynamically target specific subsets of parameters or adapt their acquisition strategy to varying inference goals at runtime.

**Related work.** A major family of ABI methods is Neural Processes (NPs) [27, 26], that learn a mapping from the observed context data points to a predictive distribution for new target points. Early NPs often employed MLP-based encoders [27, 26, 38, 11], while recent works utilize more advanced attention and transformer architectures [42, 53, 52, 19, 20, 5, 4]. Complementary to

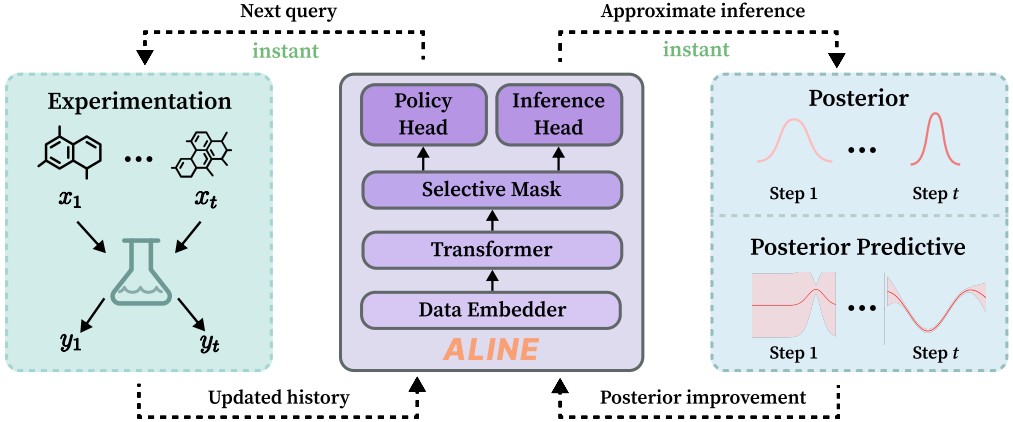

Figure 1: Conceptual workflow of ALINE, demonstrating its capability to sequentially query informative data points and perform rapid posterior or predictive inference based on the gathered data.

NPs, methods within simulation-based inference [17] focus on amortizing the posterior distribution [54, 49, 34, 60, 51, 75]. More recently, methods for amortizing both the posterior and posterior predictive distributions have been proposed [31, 59, 15]. Specifically, ACE [15] shows how to flexibly condition on diverse user-specified inference targets, a method ALINE incorporates for its own flexible inference capabilities. Building on this principle of goal-directed adaptability, ALINE advances it by integrating a learned policy that dynamically tailors the data acquisition strategy to the specified objectives. Existing amortized BED or BAL methods [23, 40, 9, 46] that learn an offline design policy do not provide real-time estimates of the posterior, unlike ALINE. Recent exceptions include methods like RL-sCEE [8] and vsOED [65], that use a variational posterior bound of EIG to provide amortized posterior inference via a separate proposal network. Compared to these methods, ALINE uses a single, unified architecture where the same model performs amortized inference for both posterior and posterior predictive distributions, and learns the flexible acquisition policy.

## 3 Amortized active learning and inference engine

**Problem setup.** We aim to develop a system that intelligently acquires a sequence of $T$ informative data points, $\mathcal{D}_T = \{(x_i, y_i)\}_{i=1}^T$, to enable accurate and rapid Bayesian inference. This system must be flexible: capable of targeting different quantities of interest, such as subsets of model parameters or future predictions. To formalize this flexibility, we introduce a *target specifier*, denoted by $\xi \in \Xi$, which defines the specific inference goal. We consider two primary types of targets: (1) **Parameter targets** ($\xi_S^\theta$) with the goal to infer a specific subset of model parameters $\theta_S$, where $S \subseteq \{1, \ldots, L\}$ is an index set of parameters of interest. For example, $\xi_{\{1,2\}}^\theta$ would target the joint posterior of $\theta_1$ and $\theta_2$, while $\xi_{\{1,\ldots,L\}}^\theta$ targets all parameters, aligning with standard BED. We define $\mathcal{S} = \{S_1, \ldots, S_{|\mathcal{S}|}\}$ as the collection of all predefined parameter index subsets the system can target. (2) **Predictive targets** ($\xi_{p_\star}^{y^\star}$), where the objective is to improve the posterior predictive distribution $p(y^\star | x^\star, \mathcal{D}_T)$ for inputs $x^\star$ drawn from a specified target input distribution $p_\star(x^\star)$. For simplicity, and following Smith et al. [67], we consider a single target distribution $p_\star(x^\star)$ in this work. The set of all target specifiers that ALINE is trained to handle is thus $\Xi = \{\xi_S^\theta\}_{S \in \mathcal{S}} \cup \{\xi_{p_\star}^{y^\star}\}$. We assume a discrete distribution $p(\xi)$ over these possible targets, reflecting the likelihood or importance of each specific goal.

To achieve both instant, informative querying and accurate inference, we propose to jointly learn an *amortized inference model* $q_\phi$ and an *acquisition policy* $\pi_\psi$ within a single, integrated architecture. Given the accumulated data history $\mathcal{D}_{t-1}$ and a specific target $\xi \in \Xi$, the policy $\pi_\psi$ selects the next query $x_t$ designed to be most informative for that target. Subsequently, the new data point $(x_t, y_t)$ is observed, and the inference model $q_\phi$ updates its estimate of the corresponding posterior or posterior predictive distribution. A conceptual workflow of ALINE is illustrated in Figure 1. In the remainder of this section, we detail the objectives for training the inference network $q_\phi$ (Section 3.1) and the acquisition policy $\pi_\psi$ (Section 3.2), discuss their practical implementation (Section 3.3), and describe the unified model architecture (Section 3.4).

### 3.1 Amortized inference

We use the inference network $q_\phi$ to provide accurate approximations of the true Bayesian posterior $p(\theta \,|\, \mathcal{D}_T)$ or posterior predictive distribution $p(y^\star \,|\, x^\star, \mathcal{D}_T)$, given the acquired data $\mathcal{D}_T$. We train $q_\phi$ via maximum-likelihood (Eq. 1). Specifically, for parameter targets $\xi = \xi_S^\theta$, our objective is:

$$\mathcal{L}_S^\theta(\phi) = -\mathbb{E}_{p(\theta)p(\mathcal{D}_T \,|\, \theta)}\left[\log q_\phi(\theta_S \,|\, \mathcal{D}_T)\right] \approx -\mathbb{E}_{p(\theta)p(\mathcal{D}_T \,|\, \theta)}\left[\sum_{l \in S} \log q_\phi(\theta_l \,|\, \mathcal{D}_T)\right], \quad (5)$$

where we adopt a diagonal or *mean field* approximation, where the joint distribution is obtained as a product of marginals $q_\phi(\theta_S \,|\, \mathcal{D}_T) \approx \prod_{l \in S} q_\phi(\theta_l \,|\, \mathcal{D}_T)$. Analogously, for predictive targets $\xi = \xi_{p_\star}^{y^\star}$, we assume a factorized likelihood over targets sampled from the target input distribution $p_\star(x^\star)$:

$$\mathcal{L}_{p_\star}^{y^\star}(\phi) = -\mathbb{E}_{p(\theta)p(\mathcal{D}_T \,|\, \theta)p_\star(x_{1:M}^\star)p(y_{1:M}^\star \,|\, x_{1:M}^\star, \theta)}\left[\log q_\phi(y_{1:M}^\star \,|\, x_{1:M}^\star, \mathcal{D}_T)\right]$$
$$\approx -\mathbb{E}_{p(\theta)p(\mathcal{D}_T \,|\, \theta)p_\star(x_{1:M}^\star)p(y_{1:M}^\star \,|\, x_{1:M}^\star, \theta)}\left[\sum_{m=1}^M \log q_\phi(y_m^\star \,|\, x_m^\star, \mathcal{D}_T)\right]. \quad (6)$$

The factorized form of these training objectives is a common scalable choice in the neural process literature [26, 52, 53, 15] and is more flexible than it might seem, as conditional marginal distributions can be extended to represent *full joints* autoregressively [53, 11, 15]. However, a full autoregressive model would require multiple forward passes to compute the reward signal for our policy at each training step, making the learning process computationally intractable. Therefore, for simplicity and tractability, within the scope of this paper we focus on the marginals, leaving the autoregressive extension to future work. Eqs. 5 and 6 form the basis for training the inference component $q_\phi$. Optimizing them minimizes the Kullback-Leibler (KL) divergence between the true target distributions (posterior or predictive) defined by the underlying generative process and the model's approximations $q_\phi$ [52]. Learning an accurate $q_\phi$ is crucial as it not only determines the quality of the final inference output but also serves as the basis for guiding the data acquisition policy, as we see next.

### 3.2 Amortized data acquisition

The quality of inference from $q_\phi$ depends critically on the informativeness of the acquired dataset $\mathcal{D}_T$. The acquisition policy $\pi_\psi$ is thus responsible for actively selecting a sequence of query-data pairs $(x_t, y_t)$ to maximize the information gained about a specific target $\xi$.

When targeting parameters $\theta_S$ (i.e., $\xi = \xi_S^\theta$), the objective is the total Expected Information Gain (sEIG$_{\theta_S}$) about $\theta_S$ over the $T$-step trajectory generated by $\pi_\psi$ (see Eq. 3):

$$\text{sEIG}_{\theta_S}(\psi) = \mathbb{E}_{p(\theta)p(\mathcal{D}_T \,|\, \pi_\psi, \theta)}\left[\log p(\theta_S \,|\, \mathcal{D}_T)\right] + H[p(\theta_S)]. \quad (7)$$

For completeness, we include a derivation of sEIG$_{\theta_S}$ in Appendix A.1 which is analogous to that of sEIG in [23]. Directly optimizing sEIG is generally intractable due to its reliance on the unknown true posterior $p(\theta_S \,|\, \mathcal{D}_T)$. We circumvent this by substituting $p(\theta_S \,|\, \mathcal{D}_T)$ with its approximation $q_\phi(\theta_S \,|\, \mathcal{D}_T)$ (from Eq. 5), yielding the tractable objective $\mathcal{J}_S^\theta$ for training $\pi_\psi$:

$$\mathcal{J}_S^\theta(\psi) := \mathbb{E}_{p(\theta)p(\mathcal{D}_T \,|\, \pi_\psi, \theta)}\left[\sum_{l \in S} \log q_\phi(\theta_l \,|\, \mathcal{D}_T)\right] + H[p(\theta_S)]. \quad (8)$$

The inference objective (Eq. 5) and this policy objective are thus coupled: $\mathcal{L}_S^\theta(\phi)$ depends on data $\mathcal{D}_T$ acquired through $\pi_\psi$, and $\mathcal{J}_S^\theta(\psi)$ depends on the inference network $q_\phi$.

Similarly, when targeting predictions for $\xi = \xi_{p_\star}^{y^\star}$, we aim to maximize information about $p(y^\star \,|\, x^\star, \mathcal{D}_T)$ for $x^\star \sim p_\star(x^\star)$. We extend the Expected Predictive Information Gain (EPIG) framework [67] to the amortized sequential setting, defining the total sEPIG:

**Proposition 1.** *The total expected predictive information gain for a design policy $\pi_\psi$ over a data trajectory of length $T$ is:*

$$\text{sEPIG}(\psi) := \mathbb{E}_{p_\star(x^\star)p(\mathcal{D}_T \,|\, \pi_\psi)}[H[p(y^\star \,|\, x^\star)] - H[p(y^\star \,|\, x^\star, \mathcal{D}_T)]]$$
$$= \mathbb{E}_{p(\theta)p(\mathcal{D}_T \,|\, \pi_\psi, \theta)p_\star(x^\star)p(y^\star \,|\, x^\star, \theta)}\left[\log p(y^\star \,|\, x^\star, \mathcal{D}_T)\right] + \mathbb{E}_{p_\star(x^\star)}[H[p(y^\star \,|\, x^\star)]].$$

This result adapts Theorem 1 in [23] for the predictive case (see Appendix A.2 for proof) and, unlike single-step EPIG (Eq. 4), considers the entire trajectory $\mathcal{D}_T$ given the policy $\pi_\psi$.

Now, similar to Eq. 8, we use the inference network $q_\phi(y^\star \mid x^\star, \mathcal{D}_T)$ to replace the true posterior predictive distribution $p(y^\star \mid x^\star, \mathcal{D}_T)$ in Proposition 1 to obtain our active learning objective:

$$\mathcal{J}_{p_\star}^{y^\star}(\psi) := \mathbb{E}_{p(\theta)p(\mathcal{D}_T \mid \pi_\psi, \theta)p_\star(x^\star)p(y^\star \mid x^\star, \theta)}\left[\log q_\phi(y^\star \mid x^\star, \mathcal{D}_T)\right] + \mathbb{E}_{p_\star(x^\star)}[H[p(y^\star \mid x^\star)]]. \quad (9)$$

Finally, the following proposition proves that our acquisition objectives, $\mathcal{J}_S^\theta$ and $\mathcal{J}_{p_\star}^{y^\star}$, are variational lower bounds on the true total information gains (sEIG$_{\theta_S}$ and sEPIG, respectively), making them principled tractable objectives for our goal. The proof is given in Appendix A.3.

**Proposition 2.** *Let the policy $\pi_\psi$ generate the trajectory $\mathcal{D}_T$. With $q_\phi(\theta_S \mid \mathcal{D}_T)$ approximating $p(\theta_S \mid \mathcal{D}_T)$, and $q_\phi(y^\star \mid x^\star, \mathcal{D}_T)$ approximating $p(y^\star \mid x^\star, \mathcal{D}_T)$, we have $\mathcal{J}_S^\theta(\psi) \leq sEIG_{\theta_S}(\psi)$ and $\mathcal{J}_{p_\star}^{y^\star}(\psi) \leq sEPIG(\psi)$. Moreover,*

$$sEIG_{\theta_S}(\psi) - \mathcal{J}_S^\theta(\psi) = \mathbb{E}_{p(\mathcal{D}_T \mid \pi_\psi)}[KL(p(\theta_S \mid \mathcal{D}_T)||q_\phi(\theta_S \mid \mathcal{D}_T))], \quad and$$

$$sEPIG(\psi) - \mathcal{J}_{p_\star}^{y^\star}(\psi) = \mathbb{E}_{p_\star(x^\star)p(\mathcal{D}_T \mid \pi_\psi)}[KL(p(y^\star \mid x^\star, \mathcal{D}_T)||q_\phi(y^\star \mid x^\star, \mathcal{D}_T))].$$

This principle of using approximate posterior (or predictive) distributions to bound information gain is foundational in Bayesian experimental design (e.g., [24]) and has been extended to sequential amortized settings [8, 65]. Maximizing these $\mathcal{J}$ objectives thus encourages policies that increase information about the targets. The tightness of these bounds is governed by the expected KL divergence between the true quantities and their approximation, with a more accurate $q_\phi$ leading to tighter bounds and a more effective training signal for the policy. Additionally, our objectives solely rely on ALINE's variational posterior, which does not require an explicit likelihood, making it naturally applicable to problems with implicit likelihoods where only forward sampling is possible.

**Data acquisition objective for ALINE.** To handle any target $\xi \in \Xi$ from a user-specified set, we unify the previously defined acquisition objectives. We define $\mathcal{J}(\psi, \xi)$ based on the type of target $\xi$:

$$\mathcal{J}(\psi, \xi) = \begin{cases} \mathcal{J}_S^\theta(\psi), & \text{if } \xi = \xi_S^\theta \\ \mathcal{J}_{p_\star}^{y^\star}(\psi), & \text{if } \xi = \xi_{p_\star}^{y^\star}. \end{cases}$$

The final objective for learning the policy network $\pi_\psi$, denoted as $\mathcal{J}^\Xi$, is the expectation of $\mathcal{J}(\psi, \xi)$ taken over the distribution of possible target specifiers $p(\xi)$: $\mathcal{J}^\Xi(\psi) = \mathbb{E}_{\xi \sim p(\xi)}[\mathcal{J}(\psi, \xi)]$.

### 3.3 Training methodology for ALINE

The policy and inference networks, $\pi_\psi$ and $q_\phi$, in ALINE are trained jointly. Training $\pi_\psi$ for sequential data acquisition over a $T$-step horizon is naturally framed as a reinforcement learning (RL) problem.

**Policy network training ($\pi_\psi$).** To guide the policy, we employ a dense, per-step reward signal $R_t$ rather than relying on a sparse reward at the end of the trajectory. This approach, common in amortized experimental design [9, 8, 37], helps stabilize and accelerate learning. The reward $R_t$ quantifies the immediate improvement in the inference quality provided by $q_\phi$ upon observing a new data point $(x_t, y_t)$, specifically concerning the current target $\xi$. It is defined based on the one-step change in the log-probabilities from our acquisition objectives (Eqs. 8 and 9):

$$R_t(\xi) = \begin{cases} \frac{1}{|S|}\sum_{l \in S}(\log q_\phi(\theta_l \mid \mathcal{D}_t) - \log q_\phi(\theta_l \mid \mathcal{D}_{t-1})), & \text{if } \xi = \xi_S^\theta \\ \frac{1}{M}\sum_{m=1}^M(\log q_\phi(y_m^\star \mid x_m^\star, \mathcal{D}_t) - \log q_\phi(y_m^\star \mid x_m^\star, \mathcal{D}_{t-1})), & \text{if } \xi = \xi_{p_\star}^{y^\star}. \end{cases} \quad (10)$$

As per common practice, for gradient stabilization we take averages (not sums) over predictions, which amounts to a constant relative rescaling of our objectives. The policy $\pi_\psi$ is then trained using a policy gradient (PG) algorithm with per-episode loss:

$$\mathcal{L}_{\text{PG}}(\psi) = -\sum_{t=1}^T \gamma^t R_t(\xi) \log \pi_\psi(x_t \mid \mathcal{D}_{t-1}, \xi), \quad (11)$$

which maximizes the expected cumulative $\gamma$-discounted reward over trajectories [68]. Gradients from this policy loss only update the policy parameters $\psi$. They are not propagated back to the inference network $q_\phi$ to ensure each component has a clear and distinct objective.

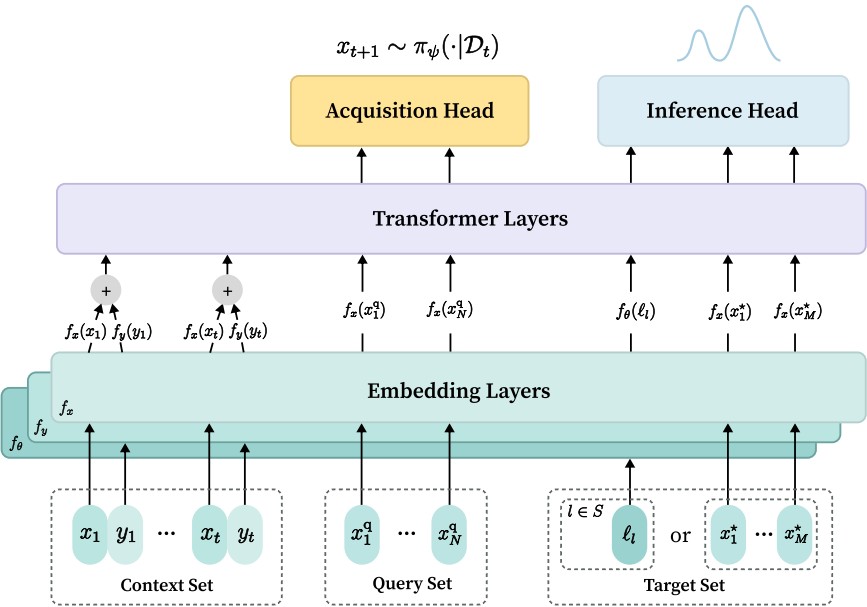

Figure 2: The ALINE architecture. The model takes historical observations (context set), candidate points (query set), and the current inference goal (target set) as inputs. These are transformed by embedding layers and subsequently by transformer layers. Finally, an acquisition head determines the next data point to query, while an inference head performs the approximate Bayesian inference.

**Inference network training ($q_\phi$).** For the per-step rewards $R_t$ to be meaningful, the inference network $q_\phi$ must provide accurate estimates of posteriors or predictive distributions at *each intermediate step* $t$ of the acquisition sequence, not just at the final step $T$. Consequently, the practical training objective for $q_\phi$, denoted $\mathcal{L}_{\mathrm{NLL}}(\phi)$, encourages this step-wise accuracy. In practice, training proceeds in episodes. For each episode: (1) A ground truth parameter set $\theta$ is sampled from the prior $p(\theta)$. (2) A target specifier $\xi$ is sampled from $p(\xi)$. (3) If the target is predictive ($\xi = \xi_{p_\star}^{y^\star}$), $M$ target inputs $\{x_m^\star\}_{m=1}^M$ are sampled from $p_\star(x^\star)$, and their corresponding true outcomes $\{y_m^\star\}_{m=1}^M$ are simulated using $\theta$, similarly as [67]. The negative log-likelihood loss $\mathcal{L}_{\mathrm{NLL}}(\phi)$ for $q_\phi$ in an episode is then computed by averaging over the $T$ acquisition steps and predictions, using the Monte Carlo estimates of the objectives defined in Eqs. 5 and 6:

$$\mathcal{L}_{\mathrm{NLL}}(\phi) \approx \begin{cases} -\frac{1}{T}\frac{1}{|S|} \sum_{t=1}^{T} \sum_{l \in S} \log q(\theta_l \mid \mathcal{D}_t), & \text{if } \xi = \xi_S^\theta \\ -\frac{1}{T}\frac{1}{M} \sum_{t=1}^{T} \sum_{m=1}^{M} \log q(y_m^\star \mid x_m^\star, \mathcal{D}_t), & \text{if } \xi = \xi_{p_\star}^{y^\star}. \end{cases} \tag{12}$$

**Joint training.** To ensure $q_\phi$ provides a reasonable reward signal early in training, we employ an initial warm-up phase where only $q_\phi$ is trained, with data acquisition $(x_t, y_t)$ guided by random actions instead of $\pi_\psi$. After the warm-up, $q_\phi$ and $\pi_\psi$ are trained jointly. A detailed step-by-step training algorithm is provided in Appendix B.1.

### 3.4 Architecture

ALINE employs a single, integrated neural architecture based on Transformer Neural Processes (TNPs) [53, 15], a network architecture that has been successfully applied to various amortized sequential decision-making settings [37, 1, 50, 39, 76]. ALINE leverages TNPs to concurrently manage historical observations, propose future queries, and condition predictions on specific, potentially varying, inference objectives. An overview of ALINE's architecture is provided in Figure 2.

The model inputs are structured into three sets. Following standard TNP-based architecture, the **context set** $\mathcal{D}_t = \{(x_i, y_i)\}_{i=1}^t$ comprises the history of observations, and the **target set** $\mathcal{T}$ contains the specific target specifier $\xi$. To facilitate active data acquisition, we incorporate a **query set** $\mathcal{Q} = \{x_n^q\}_{n=1}^N$ of candidate points. In this paper, we focus on a discrete pool-based setting for consistency, though it can be straightforwardly extended to continuous design spaces (e.g., [64]). Details regarding the embeddings of these inputs are provided in Appendix B.2.

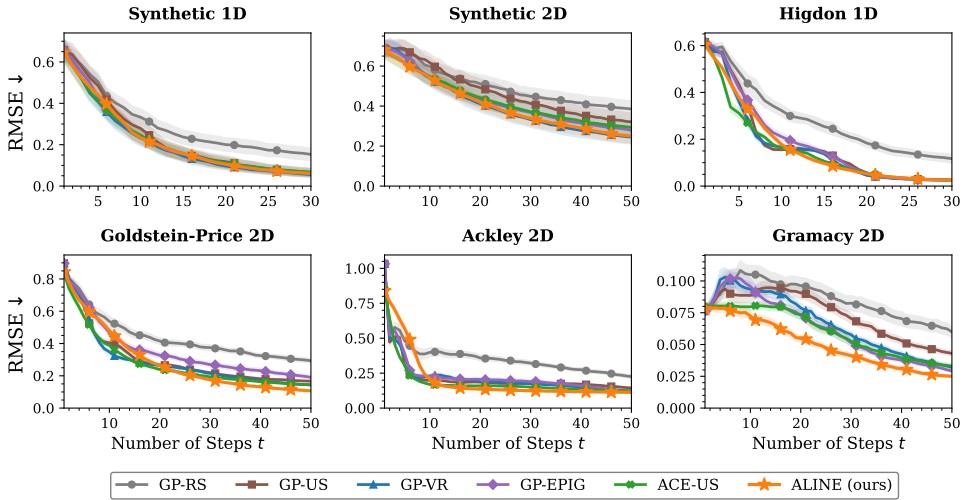

Figure 3: Predictive performance on active learning benchmark functions (RMSE ↓). Results show the mean and 95% confidence interval (CI) across 100 runs.

Standard transformer attention mechanisms process these embedded representations. Self-attention operates within the context set, capturing dependencies within $\mathcal{D}_t$. Both the query and target set then employ cross-attention to attend to the processed context set representations. To enable the policy $\pi_\psi$ to dynamically adapt its acquisition strategy based on the specific inference goal $\xi$, we introduce an additional *query-target cross-attention* mechanism to allow the query candidates to directly attend to the target set. This allows the evaluation of each candidate $x_n^q$ to be informed by its relevance to the different potential targets $\xi$. Examples of the attention mask are shown in Figure A1.

Finally, two specialized output heads operate on these processed representations. The **inference head** ($q_\phi$), following ACE [15], uses a Gaussian mixture to parameterize the approximate posteriors and posterior predictives. The **acquisition head** ($\pi_\psi$) generates a policy over the query set $\pi_\psi(x_{t+1}|\mathcal{D}_t)$, drawing on principles from policy-based design methods [37, 50]. This unified design, which leverages a shared transformer backbone with specialized heads, significantly improves parameter efficiency by avoiding the need for separate encoders for the two tasks, leading to faster training and more efficient deployment.

## 4 Experiments

We now empirically evaluate ALINE's performance in different active data acquisition and amortized inference tasks. We begin with the active learning task in Section 4.1, where we want to efficiently minimize the uncertainty over an unknown function by querying $T$ data points. Then, we test ALINE's policy on standard BED benchmarks in Section 4.2. In Section 4.3, we demonstrate the benefit of ALINE's flexible targeting feature in a psychometric modeling task [72]. Finally, to demonstrate the scalability of ALINE, we test it on a high-dimensional task of actively exploring hyperparameter performance landscapes; the results are presented in Appendix D.1. The code to reproduce our experiments is available at: `https://github.com/huangdaolang/aline`.

### 4.1 Active learning for regression and hyperparameter inference

For the active learning task, ALINE is trained on a diverse collection of fully *synthetic functions* drawn from Gaussian Process (GP) [62] priors (see Appendix C.1.1 for details). We evaluate ALINE's performance under both *in-distribution* and *out-of-distribution* settings. For the *in-distribution* setting, ALINE is evaluated on synthetic functions sampled from the same GP prior that is used during training. In the *out-of-distribution* setting, we evaluate ALINE on benchmark functions (Higdon, Goldstein-Price, Ackley, and Gramacy) *unseen* during training, to assess generalization beyond the training regime. We compare ALINE against non-amortized GP models equipped with standard acquisition functions such as Uncertainty Sampling (GP-US), Variance Reduction (GP-VR) [74], EPIG (GP-EPIG) [67], and Random Sampling (GP-RS). Additionally, we include an amortized neural

Table 2: Results on BED benchmarks. For the EIG lower bound, we report the mean ± 95% CI across 2000 runs (200 for VPCE). For deployment time, we use the mean ± 95% CI from 20 runs.

| | Location Finding | | | Constant Elasticity of Substitution | | |
|---|---|---|---|---|---|---|
| | EIG lower bound (↑) | Training time (h) | Deployment time (s) | EIG lower bound (↑) | Training time (h) | Deployment time (s) |
| Random | 5.17±0.05 | N/A | N/A | 9.05±0.26 | N/A | N/A |
| VPCE [25] | 5.25±0.22 | N/A | 146.59±0.09 | 9.40±0.27 | N/A | 788.90±1.03 |
| DAD [23] | 7.33±0.06 | 7.24 | 0.0001±0.00 | 10.77±0.15 | 13.70 | 0.0001±0.00 |
| vsOED [65] | 7.30±0.06 | 4.31 | 0.0002±0.00 | 12.12±0.18 | 0.49 | 0.0003±0.00 |
| RL-BOED [9] | 7.70±0.06 | 63.29 | 0.0003±0.00 | 14.60±0.10 | 67.28 | 0.0004±0.00 |
| ALINE (ours) | 8.91±0.04 | 21.20 | 0.03±0.00 | 13.50±0.15 | 13.29 | 0.04±0.00 |

process baseline, the Amortized Conditioning Engine (ACE) [15], paired with Uncertainty Sampling (ACE-US), to specifically evaluate the advantage of ALINE's learned acquisition policy over using a standard acquisition function with an amortized inference model. The performance metric is the root mean squared error (RMSE) of the predictions on a held-out test set.

Results for the active learning task in Figure 3 show that ALINE performs comparably to the best-performing GP-based methods for the *in-distribution* setting. Importantly, for the *out-of-distribution* setting, ALINE outperforms the baselines in 3 out of the 4 benchmark functions. These results highlight the advantage of ALINE's end-to-end learning strategy, which obviates the need for kernel specification using GPs or explicit acquisition function selection. Further evaluations on additional benchmark functions (Gramacy 1D, Branin, Three Hump Camel), visualizations of ALINE's sequential querying strategy with corresponding predictive updates, and a comparison of average inference times are provided in Appendix D.2.

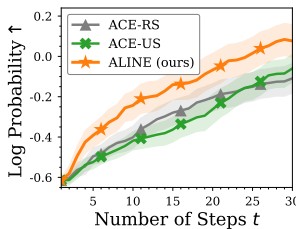

Figure 4: Hyperparameter inference performance on synthetic GP functions.

Additionally, for the *in-distribution* setting, we test ALINE's capability to infer the underlying GP's hyperparameters—without retraining, by leveraging ALINE's flexible target specification at runtime. For baselines, we use ACE-US and ACE-RS, since ACE [15] is also capable of posterior estimation. Figure 4 shows that ALINE yields higher log probabilities of the true parameter value under the estimated posterior at each step compared to the baselines. This is due to the ability to flexibly switch ALINE's acquisition strategy to parameter inference, unlike other active learning methods. We also visualize the obtained posteriors in Appendix D.2.

## 4.2 Benchmarking on Bayesian experimental design tasks

We test ALINE on two classical BED tasks: Location Finding [66] and Constant Elasticity of Substitution (CES) [3], with two- and six-dimensional design space, respectively. As baselines, we include a random design policy, a gradient-based method with variational Prior Contrastive Estimation (VPCE) [25], and three amortized BED methods: Deep Adaptive Design (DAD) [23], vsOED [65], and RL-BOED [9]. Details of the tasks and the baselines are provided in Appendix C.2.

To evaluate performance, we compute a lower bound of the total EIG, namely the sequential Prior Contrastive Estimation lower bound [23]. As shown in Table 2, ALINE surpasses all the baselines in the Location Finding task and achieves competitive performance on the CES task, outperforming most other methods, except RL-BOED. Notably, ALINE's training time is reduced as its reward is based on the internal posterior improvement and does not require a large number of contrastive samples to estimate sEIG. While ALINE's deployment time is slightly higher than MLP-based amortized methods due to the computational cost of its transformer architecture, it remains orders of magnitude faster than non-amortized approaches like VPCE. Visualizations of ALINE's inferred posterior distributions are provided in Appendix D.3.

## 4.3 Psychometric model

Our final experiment involves the psychometric modeling task [72]—a fundamental scenario in behavioral sciences, from neuroscience to clinical settings [29, 57, 73], where the goal is to infer parameters governing an observer's responses to varying stimulus intensities. The psychometric

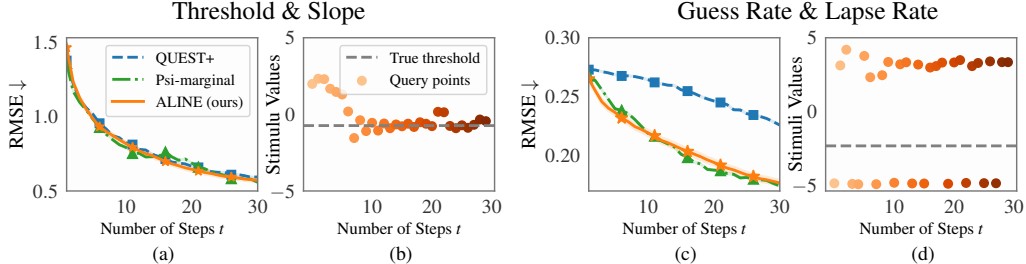

Figure 5: Results on psychometric model. RMSE (mean $\pm$ 95% CI) when targeting (a) threshold & slope and (c) guess & lapse rates, with ALINE's corresponding query strategies shown in (b) and (d).

function used here is characterized by four parameters: threshold, slope, guess rate, and lapse rate; see Appendix C.3.1 for details. Different research questions in psychophysics necessitate focusing on different parameter subsets. For instance, studies on perceptual sensitivity primarily target precise estimation of threshold and slope, while investigations into response biases or attentional phenomena might focus on the guess and lapse rates. This is where ALINE's unique flexible querying strategy can be used to target specific parameter subsets of interest.

We compare ALINE with two established adaptive psychophysical methods: QUEST+ [70], which targets all parameters simultaneously, and Psi-marginal [58], which can marginalize over nuisance parameters to focus on a specified subset, a non-amortized gold-standard method for flexible acquisition. We evaluate scenarios targeting either the threshold and slope parameters or the guess and lapse rates. Details of the baselines and the experimental setup are in Appendix C.3.2.

Figure 5 shows the results. When targeting threshold and slope (Figure 5a), which are generally easier to infer, ALINE achieves results comparable to baselines. When targeting guess and lapse rates (Figure 5c), QUEST+ performs sub-optimally as its experimental design strategy is dominated by the more readily estimable threshold and slope parameters. In contrast, both Psi-marginal and ALINE lead to significantly better inference than QUEST+ when explicitly targeting guess and lapse rates. Moreover, ALINE offers a $10\times$ speedup over these non-amortized methods (See Appendix D). We also visualize the query strategies adopted by ALINE in the two cases: when targeting threshold and slope (Figure 5b), stimuli are concentrated near the estimated threshold. Conversely, when targeting guess and lapse rates (Figure 5d), ALINE appropriately selects 'easy' stimuli at extreme values where mistakes can be more readily attributed to random behavior (governed by lapses and guesses) rather than the discriminative ability of the subject (governed by threshold and slope). To further demonstrate ALINE's runtime flexibility, we conduct two additional investigations detailed in Appendix D.4. First, we show that a single pre-trained ALINE model can dynamically switch its acquisition target mid-experiment. Second, we validate its ability to generalize to novel combinations of targets that were not seen during training, showing the effectiveness of the query-target cross-attention mechanism.

## 5 Conclusion

We introduced ALINE, a unified amortized framework that seamlessly integrates active data acquisition with Bayesian inference. ALINE dynamically adapts its strategy to target selected inference goals, offering a flexible and efficient solution for Bayesian inference and active data acquisition.

**Limitations & future work.** Currently, ALINE operates with pre-defined, fixed priors, necessitating re-training for different prior specifications. Future work could explore prior amortization [15, 71], to allow for dynamic prior conditioning. As ALINE estimates marginal posteriors, extending this to joint posterior estimation, potentially via autoregressive modeling [11, 35], is a promising direction. Note that, at deployment, we may encounter observations that differ substantially from the training data, leading to degradation in performance. This issue can potentially be tackled by combining ALINE with robust approaches such as [22, 36]. Lastly, ALINE's current architecture is tailored to fixed input dimensionalities and discrete design spaces, a common practice with TNPs [53, 50, 76, 37]. Generalizing ALINE to be dimension-agnostic [45] and to support continuous experimental designs [64] are valuable avenues for future research.

**Acknowledgements**

DH, LA and SK were supported by the Research Council of Finland (Flagship programme: Finnish Center for Artificial Intelligence FCAI, 359207). The authors acknowledge the research environment provided by ELLIS Institute Finland. LA was also supported by Research Council of Finland grants 358980 and 356498. SK was also supported by the UKRI Turing AI World-Leading Researcher Fellowship, [EP/W002973/1]. AB was supported by the Research Council of Finland grant no. 362534. The authors wish to thank Aalto Science-IT project, and CSC–IT Center for Science, Finland, for the computational and data storage resources provided.

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

# Appendix

The appendix is organized as follows:

- In Appendix A, we provide detailed derivations and proofs for the theoretical claims made regarding information gain and variational bounds.
- In Appendix B, we present the complete training algorithm and the specifics of the ALINE model.
- In Appendix C, we provide comprehensive details for each experimental setup, including task descriptions and baseline implementations.
- In Appendix D, we present additional experimental results, including further visualizations, performance on more benchmarks, and analyses of inference times.
- In Appendix E, we provide an overview of the computational resources and software dependencies for this work.

## A  Proofs of theoretical results

### A.1  Derivation of total EIG for $\theta_S$

Following Eq. 3, we can write the expression for the total expected information gain $\text{sEIG}_{\theta_S}$ about a parameter subset $\theta_S \subseteq \theta$ given data $\mathcal{D}_T$ generated under policy $\pi_\psi$ as:

$$\text{sEIG}_{\theta_S}(\psi) = H[p(\theta_S)] + \underbrace{\mathbb{E}_{p(\theta_S, \mathcal{D}_T \mid \pi_\psi)}[\log p(\theta_S \mid \mathcal{D}_T)]}_{E_1}, \tag{A1}$$

where $p(\theta_S)$ is the marginal prior for $\theta_S$, and $p(\theta_S, \mathcal{D}_T \mid \pi_\psi)$ is the joint distribution of $\theta_S$ and $\mathcal{D}_T$ under $\pi_\psi$. Now, let $\theta_R = \theta \setminus \theta_S$ be the remaining component of $\theta$ not included in $\theta_S$. Then, we can express $E_1$ from Eq. A1 as

$$\begin{aligned}
E_1 &= \int \log p(\theta_S \mid \mathcal{D}_T) p(\theta_S, \mathcal{D}_T \mid \pi_\psi) \mathrm{d}\theta_S \\
&= \int \log p(\theta_S \mid \mathcal{D}_T) \left[ \int p(\theta_S, \theta_R, \mathcal{D}_T \mid \pi_\psi) \mathrm{d}\theta_R \right] \mathrm{d}\theta_S \\
&= \int \log p(\theta_S \mid \mathcal{D}_T) \int p(\theta, \mathcal{D}_T \mid \pi_\psi) \mathrm{d}\theta \\
&= \mathbb{E}_{p(\theta, \mathcal{D}_T \mid \pi_\psi)}[\log p(\theta_S \mid \mathcal{D}_T)]. \tag{A2}
\end{aligned}$$

Plugging the above expression in Eq. A1 and noting that $p(\theta, \mathcal{D}_T \mid \pi_\psi) = p(\theta)p(\mathcal{D}_T \mid \theta, \pi_\psi)$, we arrive at the expression for $\text{sEIG}_{\theta_S}$ in Eq. 7.

### A.2  Proof of Proposition 1

**Proposition** (Proposition 1)**.** *The total expected predictive information gain for a design policy $\pi_\psi$ over a data trajectory of length $T$ is:*

$$\begin{aligned}
sEPIG(\psi) &:= \mathbb{E}_{p_\star(x^\star)p(\mathcal{D}_T \mid \pi_\psi)}[H[p(y^\star \mid x^\star)] - H[p(y^\star \mid x^\star, \mathcal{D}_T)]] \\
&= \mathbb{E}_{p(\theta)p(\mathcal{D}_T \mid \pi_\psi, \theta)p_\star(x^\star)p(y^\star \mid x^\star, \theta)} \left[ \log p(y^\star \mid x^\star, \mathcal{D}_T) \right] + \mathbb{E}_{p_\star(x^\star)}[H[p(y^\star \mid x^\star)]].
\end{aligned}$$

*Proof.* Let $p_\star(x^\star)$ be the target distribution over inputs $x^\star$ for which we want to improve predictive performance. Let $y^\star$ be the corresponding target output. The single-step EPIG for acquiring data $(x, y)$ measures the expected reduction in uncertainty (entropy) about $y^\star$ for a random target $x^\star \sim p_\star(x^\star)$:

$$\text{EPIG}(x) = \mathbb{E}_{p_\star(x^\star)p(y \mid x)}[H[p(y^\star \mid x^\star)] - H[p(y^\star \mid x^\star, x, y)]].$$

Following Theorem 1 in [23], the total EPIG, is the total expected reduction in predictive entropy from the initial prediction $p(y^\star \mid x^\star)$ to the final prediction based on the full history $p(y^\star \mid x^\star, \mathcal{D}_T)$:

$$\text{sEPIG}(\psi) = \mathbb{E}_{p_\star(x^\star)p(\mathcal{D}_T \mid \pi_\psi)}[H[p(y^\star \mid x^\star)] - H[p(y^\star \mid x^\star, \mathcal{D}_T)]] \tag{A3}$$

$$= \mathbb{E}_{p_\star(x^\star)p(\mathcal{D}_T \mid \pi_\psi)}[\mathbb{E}_{p(y^\star \mid x^\star, \mathcal{D}_T)}[\log p(y^\star \mid x^\star, \mathcal{D}_T)]] + \mathbb{E}_{p_\star(x^\star)}[H[p(y^\star \mid x^\star)]] \tag{A4}$$

$$= \mathbb{E}_{p_\star(x^\star)p(\mathcal{D}_T, y^\star \mid \pi_\psi, x^\star)}[\log p(y^\star \mid x^\star, \mathcal{D}_T)] + \mathbb{E}_{p_\star(x^\star)}[H[p(y^\star \mid x^\star)]]. \tag{A5}$$

Here, Eq. A3 follows from conditioning EPIG on the entire trajectory $\mathcal{D}_T$ instead of a single data point $y$, Eq. A4 follows from the definition of entropy $H[\cdot]$, and Eq. A5 follows from noting that $p(\mathcal{D}_T, y^\star \mid \pi_\psi, x^\star) = p(\mathcal{D}_T \mid \pi_\psi)p(y^\star \mid x^\star, \mathcal{D}_T)$. Next, we combine the expectations and express the joint distribution $p(\mathcal{D}_T, y^\star \mid \pi_\psi, x^\star) = \int p(\theta)p(\mathcal{D}_T \mid \pi_\psi, \theta)p(y^\star \mid x^\star, \theta)d\theta$, where, following [67], we assume conditional independence between $\mathcal{D}_T$ and $y^\star$ given $\theta$. This yields:

$$\text{sEPIG}(\psi) = \mathbb{E}_{p(\theta)p(\mathcal{D}_T \mid \pi_\psi, \theta)p_\star(x^\star)p(y^\star \mid x^\star, \theta)} \left[\log p(y^\star \mid x^\star, \mathcal{D}_T)\right] + \mathbb{E}_{p_\star(x^\star)}[H[p(y^\star \mid x^\star)]],$$

which completes our proof. $\qquad\square$

### A.3 Proof of Proposition 2

**Proposition** (Proposition 2). *Let the policy $\pi_\psi$ generate the trajectory $\mathcal{D}_T$. With $q_\phi(\theta_S \mid \mathcal{D}_T)$ approximating $p(\theta_S \mid \mathcal{D}_T)$, and $q_\phi(y^\star \mid x^\star, \mathcal{D}_T)$ approximating $p(y^\star \mid x^\star, \mathcal{D}_T)$, we have $\mathcal{J}_S^\theta(\psi) \leq sEIG_{\theta_S}(\psi)$ and $\mathcal{J}_{p_\star}^{y^\star}(\psi) \leq sEPIG(\psi)$. Moreover,*

$$sEIG_{\theta_S}(\psi) - \mathcal{J}_S^\theta(\psi) = \mathbb{E}_{p(\mathcal{D}_T \mid \pi_\psi)}[KL(p(\theta_S \mid \mathcal{D}_T)||q_\phi(\theta_S \mid \mathcal{D}_T))], \quad \text{and} \tag{A6}$$

$$sEPIG(\psi) - \mathcal{J}_{p_\star}^{y^\star}(\psi) = \mathbb{E}_{p_\star(x^\star)p(\mathcal{D}_T \mid \pi_\psi)}[KL(p(y^\star \mid x^\star, \mathcal{D}_T)||q_\phi(y^\star \mid x^\star, \mathcal{D}_T))]. \tag{A7}$$

*Proof.* Using the expressions for $\text{sEIG}_{\theta_S}$ and $\mathcal{J}_S^\theta$ from Eq. 7 and Eq. 8, respectively, and noting that $\log q_\phi(\theta_S \mid \mathcal{D}_T) = \sum_{l \in S} \log q_\phi(\theta_l \mid \mathcal{D}_T)$, we can write the expression for $\text{sEIG}_{\theta_S}(\psi) - \mathcal{J}_S^\theta(\psi)$ as:

$$\text{sEIG}_{\theta_S}(\psi) - \mathcal{J}_S^\theta(\psi) = \mathbb{E}_{p(\theta)p(\mathcal{D}_T \mid \pi_\psi, \theta)}[\log p(\theta_S \mid \mathcal{D}_T)] - \mathbb{E}_{p(\theta)p(\mathcal{D}_T \mid \pi_\psi, \theta)}[\log q_\phi(\theta_S \mid \mathcal{D}_T)]$$

$$= \mathbb{E}_{p(\mathcal{D}_T, \theta \mid \pi_\psi)}\left[\log \frac{p(\theta_S \mid \mathcal{D}_T)}{q_\phi(\theta_S \mid \mathcal{D}_T)}\right] \tag{A8}$$

$$= \mathbb{E}_{p(\mathcal{D}_T, \theta_S \mid \pi_\psi)}\left[\log \frac{p(\theta_S \mid \mathcal{D}_T)}{q_\phi(\theta_S \mid \mathcal{D}_T)}\right] \tag{A9}$$

$$= \mathbb{E}_{p(\mathcal{D}_T \mid \pi_\psi)}\left[\mathbb{E}_{p(\theta_S \mid \mathcal{D}_T)}\left[\log \frac{p(\theta_S \mid \mathcal{D}_T)}{q_\phi(\theta_S \mid \mathcal{D}_T)}\right]\right] \tag{A10}$$

$$= \mathbb{E}_{p(\mathcal{D}_T \mid \pi_\psi)}[KL(p(\theta_S \mid \mathcal{D}_T)||q_\phi(\theta_S \mid \mathcal{D}_T))]. \tag{A11}$$

Here, Eq. A8 follows from the fact $p(\theta)p(\mathcal{D}_T \mid \pi_\psi, \theta) = p(\mathcal{D}_T, \theta \mid \pi_\psi)$, Eq. A9 follows from Eq. A2, Eq. A10 follows from the fact that $p(\mathcal{D}_T, \theta_S \mid \pi_\psi) = p(\mathcal{D}_T \mid \pi_\psi)p(\theta_S \mid \mathcal{D}_T)$, and Eq. A11 follows from the definition of KL divergence.

Since the KL divergence is always non-negative ($KL(P||Q) \geq 0$), its expectation over trajectories $p(\mathcal{D}_T \mid \pi_\psi)$ must also be non-negative. Therefore:

$$\mathcal{J}_S^\theta(\psi) \leq \text{sEIG}_{\theta_S}(\psi). \tag{A12}$$

Now, we consider the difference between $\text{sEPIG}(\psi)$ and $\mathcal{J}_{p_\star}^{y^\star}(\psi)$:

$$\text{sEPIG}(\psi) - \mathcal{J}_{p_\star}^{y^\star}(\psi) = \mathbb{E}_{p(\theta)p(\mathcal{D}_T \mid \pi_\psi, \theta)p_\star(x^\star)p(y^\star \mid x^\star, \theta)}\left[\log \frac{p(y^\star \mid x^\star, \mathcal{D}_T)}{q_\phi(y^\star \mid x^\star, \mathcal{D}_T)}\right]$$
$$= \mathbb{E}_{p_\star(x^\star)p(\mathcal{D}_T \mid \pi_\psi)}\left[\mathbb{E}_{p(y^\star \mid x^\star, \mathcal{D}_T)}\left[\log \frac{p(y^\star \mid x^\star, \mathcal{D}_T)}{q_\phi(y^\star \mid x^\star, \mathcal{D}_T)}\right]\right]. \tag{A13}$$

Similar to the previous case, the inner expectation is the definition of the KL divergence between the true posterior predictive $p(y^\star \mid x^\star, \mathcal{D}_T)$ and the variational approximation $q_\phi(y^\star \mid x^\star, \mathcal{D}_T)$:

$$\text{sEPIG}(\psi) - \mathcal{J}_{p_\star}^{y^\star}(\psi) = \mathbb{E}_{p_\star(x^\star)p(\mathcal{D}_T \mid \pi_\psi)}[KL(p(y^\star \mid x^\star, \mathcal{D}_T)||q_\phi(y^\star \mid x^\star, \mathcal{D}_T))]. \tag{A14}$$

Since the KL divergence is always non-negative, therefore:

$$\mathcal{J}_{p_\star}^{y^\star}(\psi) \leq \text{sEPIG}(\psi), \tag{A15}$$

which completes the proof. $\qquad\square$

# B    Further details on ALINE

## B.1    Training algorithm

---

**Algorithm 1** ALINE Training Procedure

---

1: **Input:** Prior $p(\theta)$, likelihood $p(y \mid x, \theta)$, target distribution $p(\xi)$, query horizon $T$, total training
    episodes $E_{max}$, warm-up episodes $E_{warm}$.
2: **Output:** Trained ALINE model $(q_\phi, \pi_\psi)$.
3: **for** epoch = 1 to $E_{max}$ **do**
4:      Sample parameters $\theta \sim p(\theta)$.
5:      Sample target specifier set $\xi \sim p(\xi)$ and corresponding targets $\theta_S$ or $\{x_m^\star, y_m^\star\}_{m=1}^M$.
6:      Initialize candidate query set $\mathcal{Q}$.
7:      **for** $t = 1$ to $T$ **do**
8:          **if** epoch $\leq E_{warm}$ **then**
9:              Select next query $x_t$ uniformly at random from $\mathcal{Q}$.
10:         **else**
11:             Select next query $x_t \sim \pi_\psi(\cdot \mid \mathcal{D}_{t-1}, \xi)$ from $\mathcal{Q}$.
12:         **end if**
13:         Sample outcome $y_t \sim p(y \mid x_t, \theta)$.
14:         Update history $\mathcal{D}_t \leftarrow \mathcal{D}_{t-1} \cup \{(x_t, y_t)\}$.
15:         Update query set $\mathcal{Q} \leftarrow \mathcal{Q} \setminus \{x_t\}$.
16:         **if** epoch $\leq E_{warm}$ **then**
17:             $\mathcal{L} = \mathcal{L}_{\text{NLL}}$ (Eq. 12)
18:         **else**
19:             Calculate reward $R_t$ (Eq. 10)
20:             $\mathcal{L} = \mathcal{L}_{\text{NLL}}$ (Eq. 12) $+\mathcal{L}_{\text{PG}}$ (Eq. 11)
21:         **end if**
22:         Update ALINE using $\mathcal{L}$.
23:      **end for**
24: **end for**

---

## B.2    Architecture and training details

In ALINE, the data is first processed by different embedding layers. Inputs (context $x_i$, query candidates $x_n^q$, target locations $x_k^\star$) are passed through a shared nonlinear embedder $f_x$. Observed outcomes $y_i$ are embedded using a separate embedder $f_y$. For discrete parameters, we assign a unique indicator $\ell_l$ to each parameter $\theta_l$, which is then associated with a unique, learnable embedding vector, denoted as $f_\theta(\ell_l)$. We compute the final context embedding by summing the outputs of the respective embedders: $E^{\mathcal{D}_t} = \{(f_x(x_i) + f_y(y_i))\}_{i=1}^t$. Query and target sets are embedded as $E^{\mathcal{Q}} = \{(f_x(x_n^q))\}_{n=1}^N$ and $E^{\mathcal{T}}$ (either $\{(f_x(x_m^\star))\}_{m=1}^M$ or $\{f_\theta(\ell_l)\}_{l \in S}$). Both $f_x$ and $f_y$ are MLPs consisting of an initial linear layer, followed by a ReLU activation function, and a final linear layer. For all our experiments, the embedders use a feedforward dimension of 128 and project inputs to an embedding dimension of 32.

The core of our architecture is a transformer network. We employ a configuration with 3 transformer layers, each equipped with 4 attention heads. The feedforward networks within each transformer layer have a dimension of 128. The model's internal embedding dimension, consistent across the transformer layers and the output of the initial embedding layers, is 32. These transformer layers process the embedded representations of the context, query, and target sets. The interactions between these sets are governed by specific attention masks, visually detailed in Figure A1, where a shaded element indicates that the token corresponding to its row is permitted to attend to the token corresponding to its column.

ALINE has two specialized output heads. The inference head, responsible for approximating posteriors and posterior predictive distributions, parameterizes a Gaussian Mixture Model (GMM) with 10 components. The embeddings corresponding to the inference targets are processed by 10 separate MLPs, one for each GMM component. Each MLP outputs parameters for its component: a mixture weight, a mean, and a standard deviation. The standard deviations are passed through a Softplus

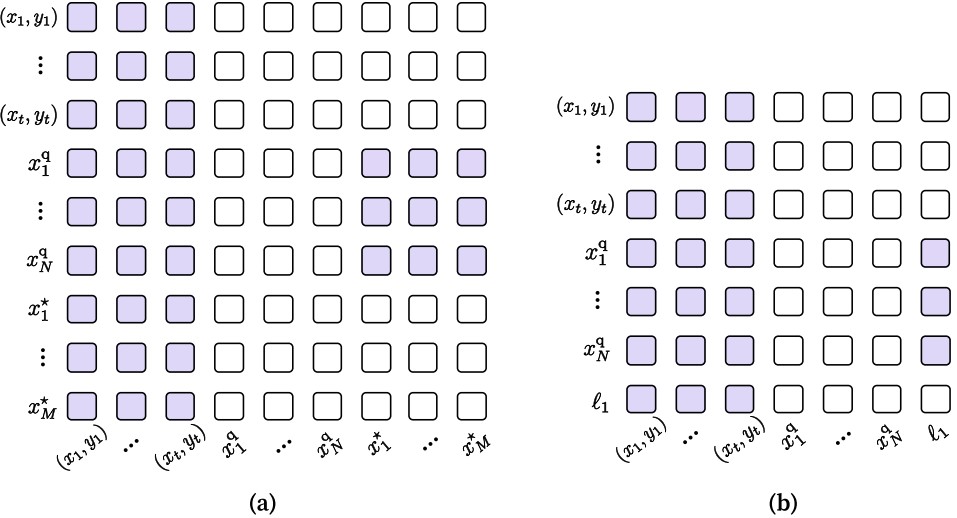

Figure A1: Example attention masks in ALINE's transformer architecture. (a) Mask for a predictive target $\xi = \xi_{p_\star}^{y^\star}$ (b) Mask for a parameter target $\xi = \xi_{\{1\}}^{\theta}$. Shaded squares indicate allowed attention.

activation function to ensure positivity, and the mixture weights are normalized using a Softmax function. The policy head, which generates a probability distribution over the candidate query points, is a 2-layer MLP with a feedforward dimension of 128. Its output is passed through a Softmax function to ensure that the probabilities of all actions sum to unity. The architecture of ALINE is shown in Figure 2.

ALINE is trained using the AdamW optimizer with a weight decay of 0.01. The initial learning rate is set to 0.001 and decays according to a cosine annealing schedule.

## C Experimental details

This section provides details for the experimental setups. Appendix C.1 outlines the specifics for the active learning experiments in Section 4.1, including the synthetic function sampling procedures (Appendix C.1.1), implementation details for baseline methods (Appendix C.1.2), and training and evaluation details for these tasks (Appendix C.1.3). Next, in Appendix C.2 we describe the details of BED tasks, including the task descriptions (Appendix C.2.1), implementation of the baselines (Appendix C.2.2), and the training and evaluation details (Appendix C.2.3). Lastly, Appendix C.3 contains the specifics of the psychometric modeling experiments, detailing the psychometric function we use (Appendix C.3.1) and the setup for the experimental comparisons (Appendix C.3.2).

### C.1 Active learning for regression and hyperparameter inference

#### C.1.1 Synthetic functions sampling procedure

For active learning tasks, ALINE is trained exclusively on synthetically generated Gaussian Process (GP) functions. The procedure for generating these functions is as follows. First, the hyperparameters of the GP kernels, namely the output scale and lengthscale(s), are sampled from their respective prior distributions. For multi-dimensional input spaces ($d_x > 1$), there is a $p_{\text{iso}} = 0.5$ probability that an isotropic kernel is used, meaning that all input dimensions share a common lengthscale. Otherwise, an anisotropic kernel is employed, with a distinct lengthscale sampled for each input dimension. Subsequently, a kernel function is chosen randomly from a pre-defined set, with each kernel having a uniform probability of selection. In our experiments, we utilize the Radial Basis Function (RBF), Matérn 3/2, and Matérn 5/2 kernels.

The kernel's output scale is sampled uniformly from the interval $U(0.1, 1)$. The lengthscale(s) are sampled from $U(0.1, 2) \times \sqrt{d_x}$. Input data points $x$ are sampled uniformly within the range $[-5, 5]$ for each dimension. Finally, Gaussian noise with a fixed standard deviation of 0.01 is added to

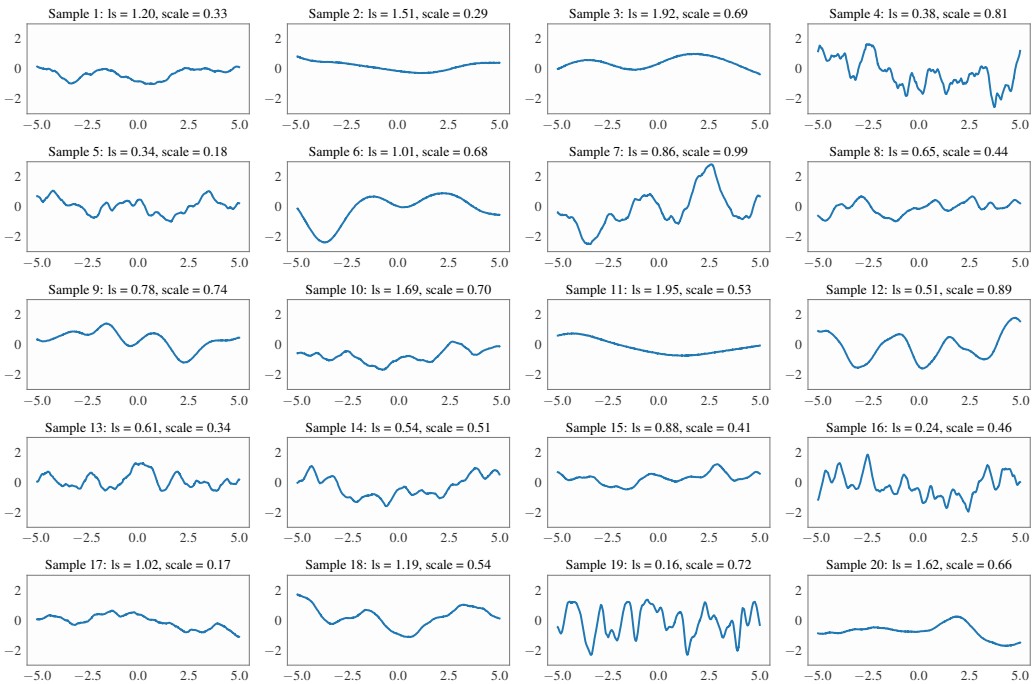

Figure A2: Examples of randomly sampled 1D synthetic GP functions used to train ALINE.

the true function output $y$ for each sampled data point. Figure A2 illustrates some examples of the synthetic GP functions generated using this procedure.

### C.1.2 Details of acquisition functions

We compare ALINE with four commonly used AL acquisition functions. For **Random Sampling (RS)**, we randomly select one point from the candidate pool as the next query point.

**Uncertainty Sampling (US)** is a simple and widely used AL acquisition strategy that prioritizes points where the model is most uncertain about its prediction:

$$\text{US}(x) = \sqrt{\mathbb{V}[y \mid x, \mathcal{D}]}, \tag{A16}$$

where $\mathbb{V}[y \mid x, \mathcal{D}]$ is the predictive variance at $x$ given the current training data $\mathcal{D}$.

**Variance Reduction (VR)** [74] aims to select a candidate point that is expected to maximally reduce the predictive variance over a pre-defined test set $\{x_m^\star\}_{m=1}^M$, which is defined as:

$$\text{VR}(x) = \frac{\sum_{m=1}^M \left(\text{Cov}_{\text{post}}(x^\star, x)\right)^2}{\mathbb{V}[y \mid x, \mathcal{D}]}, \tag{A17}$$

$\text{Cov}_{\text{post}}(x^\star, x)$ is the posterior covariance between the latent function values at $x^\star$ and $x$, given the history $\mathcal{D} = \{(X_{\text{train}}, y_{\text{train}})\}$, where $X_{\text{train}}$ comprises all currently observed inputs with $y_{\text{train}}$ being their corresponding outputs. It is computed as:

$$\text{Cov}_{\text{post}}(x^\star, x) = k(x^\star, x) - k(x^\star, X_{\text{train}})(K_{\text{train}} + \alpha I)^{-1} k(X_{\text{train}}, x). \tag{A18}$$

Here, $k(\cdot, \cdot)$ is the GP kernel function, $K_{\text{train}} = k(X_{\text{train}}, X_{\text{train}})$, and $\alpha$ is the noise variance.

**Expected Predictive Information Gain (EPIG)** [67] measures the expected reduction in predictive uncertainty on a target input distribution $p_\star(x^\star)$. Following Smith et al. [67], for a Gaussian predictive distribution, the EPIG for a candidate point can be expressed as:

$$\text{EPIG}(x) = \mathbb{E}_{p_\star(x^\star)}\left[\frac{1}{2} \log \frac{\mathbb{V}[y \mid x, \mathcal{D}]\mathbb{V}[y^\star \mid x^\star, \mathcal{D}]}{\mathbb{V}[y \mid x, \mathcal{D}]\mathbb{V}[y^\star \mid x^\star, \mathcal{D}] - \text{Cov}_{\text{post}}(x^\star, x)^2}\right]. \tag{A19}$$

In practice, we approximate it by averaging over $m$ sampled test points:

$$\text{EPIG}(x) \approx \frac{1}{2M} \sum_{m=1}^{M} \log \frac{\mathbb{V}[y \mid x, \mathcal{D}]\mathbb{V}[y_m^\star \mid x_m^\star, \mathcal{D}]}{\mathbb{V}[y \mid x, \mathcal{D}]\mathbb{V}[y_m^\star \mid x_m^\star, \mathcal{D}] - \text{Cov}_{\text{post}}(x_m^\star, x)^2}. \tag{A20}$$

### C.1.3 Training and evaluation details

For both 1D and 2D input scenarios, ALINE is trained for $2 \cdot 10^5$ epochs using a batch size of 200. The discount factor $\gamma$ for the policy gradient loss is set to 1. For the GP-based baselines, we utilized Gaussian Process Regressors implemented via the `scikit-learn` library [56]. The hyperparameters of the GP models are optimized at each step. For the ACE baseline [15], we use a transformer architecture and an inference head design consistent with our ALINE model.

All active learning experiments are evaluated with a candidate query pool consisting of 500 points. Each experimental run commenced with an initial context set consisting of a single data point. The target set size for predictive tasks is set to 100.

## C.2 Benchmarking on Bayesian experimental design tasks

### C.2.1 Task descriptions

**Location Finding** [66] is a benchmark problem commonly used in BED literature [24, 40, 9, 41]. The objective is to infer the unknown positions of $K$ hidden sources, $\theta = \{\theta_k \in \mathbb{R}^d\}_{k=1}^K$, by strategically selecting a sequence of observation locations, $x \in \mathbb{R}^d$. Each source emits a signal whose intensity attenuates with distance following an inverse-square law. The total signal intensity at an observation location $x$ is given by the superposition of signals from all sources:

$$\mu(\theta, x) = b + \sum_{k=1}^{K} \frac{\alpha_k}{m + \|\theta_k - x\|^2}, \tag{A21}$$

where $\alpha_k$ are known source strength constants, and $b, m > 0$ are constants controlling the background level and maximum signal intensity, respectively. In this experiment, we use $K = 1$, $d = 2$, $\alpha_k = 1$, $b = 0.1$ and $m = 10^{-4}$, and the prior distribution over each component of a source's location $\theta_k = (\theta_{k,1}, ..., \theta_{k,d})$ is uniform over the interval $[0, 1]$.

The observation is modeled as the log-transformed total intensity corrupted by Gaussian noise:

$$\log y \mid \theta, x \sim \mathcal{N}(\log \mu(\theta, x), \sigma^2), \tag{A22}$$

where we use $\sigma = 0.5$ in our experiments.

**Constant Elasticity of Substitution (CES)** [3] considers a behavioral economics problem in which a participant compares two baskets of goods and rates the subjective difference in utility between the baskets on a sliding scale from 0 to 1. The utility of a basket $z$, consisting of $K$ goods with different values, is characterized by latent parameters $\theta = (\rho, \boldsymbol{\alpha}, u)$. The design problem is to select pairs of baskets, $x = (z, z') \in [0, 100]^{2K}$, to infer the participant's latent utility parameters.

The utility of a basket $z$ is defined using the constant elasticity of substitution function, as:

$$U(z) = \left( \sum_{i=1}^{K} z_i^\rho \alpha_i \right)^{\frac{1}{\rho}}. \tag{A23}$$

The prior of the latent parameters is specified as:

$$\begin{aligned} \rho &\sim \text{Beta}(1, 1) \\ \boldsymbol{\alpha} &\sim \text{Dirichlet}(\mathbf{1}_K) \\ \log u &\sim \mathcal{N}(1, 3^2). \end{aligned} \tag{A24}$$

The subjective utility difference between two baskets is modeled as follows:

$$\begin{aligned} \eta &\sim \mathcal{N}\left( u \cdot (U(z) - U(z')), u^2 \cdot \tau^2 \cdot (1 + \|z - z'\|)^2 \right) \\ y &= \text{clip}(\text{sigmoid}(\eta), \epsilon, 1 - \epsilon). \end{aligned} \tag{A25}$$

In this experiment, we choose $K = 3$, $\tau = 0.005$ and $\epsilon = 2^{-22}$.

### C.2.2 Implementation details of baselines

We compare ALINE with four baseline methods. For **Random Design** policy, we randomly sample a design from the design space using a uniform distribution.

**VPCE** [25] iteratively infers the posterior through variational inference and maximizes the myopic Prior Contrastive Estimation (PCE) lower bound by gradient descent with respect to the experimental design. The hyperparameters used in the experiments are given in Table A1.

Table A1: Additional hyperparameters used in VPCE [25].

| Parameter | Location Finding | CES |
|---|---|---|
| VI gradient steps | 1000 | 1000 |
| VI learning rate | $10^{-3}$ | $10^{-3}$ |
| Design gradient steps | 2500 | 2500 |
| Design learning rate | $10^{-3}$ | $10^{-3}$ |
| Contrastive samples $L$ | 500 | 10 |
| Expectation samples | 500 | 10 |

**Deep Adaptive Design (DAD)** [23] learns an amortized design policy guided by sPCE lower bound. For a design policy $\pi$, and $L \geq 0$ contrastive samples, sPCE over a sequence of $T$ experiments is defined as:

$$\mathcal{L}_T(\pi, L) = \mathbb{E}_{p(\theta_0, \mathcal{D}_T | \pi) p(\theta_{1:L})} \left[ \log \frac{p(\mathcal{D}_T | \theta_0, \pi)}{\frac{1}{L+1} \sum_{\ell=0}^{L} p(\mathcal{D}_T | \theta_\ell, \pi)} \right], \tag{A26}$$

where the contrastive samples $\theta_{1:L}$ are drawn independently from the prior $p(\theta)$. The bound becomes tight as $L \to \infty$, with a convergence rate of $\mathcal{O}(L^{-1})$.

The design network comprises an MLP encoder that encodes historical data into a fixed-dimensional representation, and an MLP emitter that proposes the next design point. The encoder processes the concatenated design-observation pairs from history and aggregates their representations through a pooling operation.

Following the work of Foster et al. [23], the encoder network consists of two fully connected layers with 128 and 16 units with ReLU activation applied to the hidden layer. The emitter is implemented as a fully connected layer that maps the pooled representation to the design space. The policy is trained using the Adam optimizer with an initial learning rate of $5 \cdot 10^{-5}$, $\beta = (0.8, 0.998)$, and an exponentially decaying learning rate, reduced by a factor of $\gamma = 0.98$ every 1000 epochs. In the Location Finding task, the model is trained for $10^5$ epochs, and $10^4$ contrastive samples are utilized in each training step for the estimation of the sPCE lower bound. Note that, for the CES task, we applied several adjustments, including normalizing the input, applying the Sigmoid and Softplus transformations to the output before mapping it to the design space, increasing the depth of the network, and initializing weights using Xavier initialization [32]. However, DAD failed to converge during training in our experiments. Therefore, we report the results provided by Blau et al. [9].

**vsOED** [65] is an amortized BED method that employs an actor-critic reinforcement learning framework. It utilizes separate networks for the actor (policy), the critic (value function), and the variational posterior approximation. The reward signal for training the policy is the incremental improvement in the log-probability of the ground-truth parameters under the variational posterior, which is estimated at each step of the experiment. Following the original implementation, a distinct posterior network is trained for each design stage, while the actor and critic share a common backbone. For our implementation, the hidden layers of all networks are 3-layer MLPs with 256 units and ReLU activations. The posterior network outputs the parameters for an 8-component Gaussian Mixture Model (GMM). The input to the actor and critic networks is the zero-padded history of design-observation pairs, concatenated with a one-hot encoding of the current time step. We train for $10^4$ epochs with a batch size of $10^4$ and a $10^6$-sized replay buffer. The learning rate starts at $10^{-3}$ with a 0.9999 exponential decay per epoch, and the discount factor is 0.9. To encourage exploration for the deterministic policy, Gaussian noise is added during training; the initial noise scale is 0.5

for the Location Finding task and 5.0 for the CES task, with decay rates of 0.9999 and 0.9998, respectively.

**RL-BOED** [9] frames the design policy optimization as a Markov Decision Process (MDP) and employs reinforcement learning to learn the design policy. It utilizes a stepwise reward function to estimate the marginal contribution of the $t$-th experiment to the sEIG.

The design network shares a similar backbone architecture to that of DAD, with the exception that the deterministic output of the emitter is replaced by a Tanh-Gaussian distribution. The encoder comprises two fully connected layers with 128 units and ReLU activation, followed by an output layer with 64 units and no activation. Training is conducted using Randomized Ensembled Double Q-learning (REDQ) [16], the full configurations are reported in Table A2.

Table A2: Additional hyperparameters used in RL-BOED [9].

| Parameter | Location Finding | CES |
|---|---|---|
| Critics | 2 | 2 |
| Random subsets | 2 | 2 |
| Contrastive samples $L$ | $10^5$ | $10^5$ |
| Training epochs | $10^5$ | $5 \cdot 10^4$ |
| Discount factor $\gamma$ | 0.9 | 0.9 |
| Target update rate | $10^{-3}$ | $5 \cdot 10^{-3}$ |
| Policy learning rate | $10^{-4}$ | $3 \cdot 10^{-4}$ |
| Critic learning rate | $3 \cdot 10^{-4}$ | $3 \cdot 10^{-4}$ |
| Buffer size | $10^7$ | $10^6$ |

### C.2.3 Training and evaluation details

In the Location Finding task, the number of sequential design steps, $T$, is set to 30. For all evaluated methods, the sPCE lower bound is estimated using $L = 10^6$ contrastive samples. The ALINE is trained over $10^5$ epochs with a batch size of 200. The discount factor $\gamma$ for the policy gradient loss is set to 1. During the evaluation phase, the query set consists of 2000 points, which are drawn uniformly from the defined design space. For the CES task, each experimental run consists of $T = 10$ design steps. The sPCE lower bound is estimated using $L = 10^7$ contrastive samples. The ALINE is trained for $2 \cdot 10^5$ epochs with a batch size of 200, and we use 2000 for the query set size.

### C.3 Psychometric model

### C.3.1 Model description

In this experiment, we use a four-parameter psychometric function with the following parameterization:

$$\pi(x) = \theta_3 \cdot \theta_4 + (1 - \theta_4)F(\frac{x - \theta_1}{\theta_2}),$$

where:

- $\theta_1$ (threshold): The stimulus intensity at which the probability of a positive response reaches a specific criterion. It represents the location of the psychometric curve. We use a uniform prior $U[-3, 3]$ for $\theta_1$.

- $\theta_2$ (slope): Describes the steepness of the psychometric function. Smaller values of $\theta_2$ indicate a sharper transition, reflecting higher sensitivity around the threshold. We use a uniform prior $U[0.1, 2]$ for $\theta_2$.

- $\theta_3$ (guess rate): The baseline response probability for stimuli far below the threshold, reflecting responses made by guessing. We use a uniform prior $U[0.1, 0.9]$ for $\theta_3$.

- $\theta_4$ (lapse rate): The rate at which the observer makes errors independent of stimulus intensity, representing an upper asymptote on performance below 1. We use a uniform prior $U[0, 0.5]$ for $\theta_4$.

We employ a Gumbel-type internal link function $F = 1 - e^{-10^z}$ where $z = \frac{x-\theta_1}{\theta_2}$. Lastly, a binary response $y$ is simulated from the psychometric function $\pi(x)$ using a Bernoulli distribution with probability of success $p = \pi(x)$.

### C.3.2 Experimental details

We compare ALINE against two established Bayesian adaptive methods:

- QUEST+ [70]: QUEST+ is an adaptive psychometric procedure that aims to find the stimulus that maximizes the expected information gain about the parameters of the psychometric function, or equivalently, minimizes the expected entropy of the posterior distribution over the parameters. It typically operates on a discrete grid of possible parameter values and selects stimuli to reduce uncertainty over this entire joint parameter space. In our experiments, QUEST+ is configured to infer all four parameters simultaneously.

- Psi-marginal [58]: The Psi-marginal method is an extension of the psi method [43] that allows for efficient inference by marginalizing over nuisance parameters. When specific parameters are designated as targets of interest, Psi-marginal optimizes stimulus selection to maximize information gain specifically for these target parameters, effectively treating the others as nuisance variables. This makes it highly efficient when only a subset of parameters is critical.

For each simulated experiment, true underlying parameters are sampled from their prior distributions. Stimulus values $x$ are selected from a discrete set of size 200 drawn uniformly from the range $[-5, 5]$.

## D   Additional experimental results

### D.1   Active Exploration of High-Dimensional Hyperparameter Landscapes

To demonstrate ALINE's utility on complex, high-dimensional tasks, we conduct a new set of experiments on actively exploring hyperparameter performance landscapes. This experiment aims to efficiently characterize a machine learning model's overall behavior on a new task, allowing practitioners to quickly assess a model family's viability or understand its sensitivities. The task is to actively query a small number of hyperparameter configurations to build a surrogate model that accurately predicts performance for a larger, held-out set of target configurations. We use high-dimensional, real-world tasks from the HPO-B benchmark [2], evaluating on `rpart` (6D), `svm` (8D), `ranger` (9D), and `xgboost` (16D) datasets. ALINE is trained on their multiple pre-defined training sets. We then evaluate its performance, alongside non-amortized GP-based baselines and an amortized surrogate baseline (ACE-US), on the benchmark's held-out and entirely unseen test sets.

Table A3 shows the RMSE results after 30 steps, averaged across all test tasks for each dataset. First, both amortized methods, ALINE and ACE-US, significantly outperform all non-amortized GP-based baselines across all tasks. This highlights the power of meta-learning in this domain. GP-based methods must learn each new performance landscape from scratch, which is highly inefficient in high dimensions. In contrast, both ALINE and ACE-US are pre-trained on hundreds of related tasks, and their Transformer architectures meta-learn the structural patterns common to these landscapes. This shared prior knowledge allows them to make far more accurate predictions from sparse data. Second, while ACE-US performs strongly due to its amortized nature, ALINE consistently achieves the best or joint-best performance. This demonstrates the additional, crucial benefit of our core contribution: the learned acquisition policy. ACE-US relies on a standard heuristic, whereas ALINE's policy is trained end-to-end to learn how to optimally explore the landscape, leading to more informative queries and ultimately a more accurate final surrogate model.

### D.2   Active learning for regression and hyperparameter inference

**AL results on more benchmark functions.**   To further assess ALINE, we present performance evaluations on an additional set of active learning benchmark functions, see Figure A3. The results on Gramacy and Branin show that we are on par with the GP baselines. For the Three Hump Camel, we see both ALINE and ACE-US showing reduced accuracy. This is because the function's output value range extends beyond that of the GP functions used during pre-training. This highlights a potential

Table A3: RMSE ($\downarrow$) on the HPO-B benchmark after 30 active queries. Results show the mean and 95% CI over all test tasks for each dataset.

|              | GP-RS         | GP-US         | GP-VR         | GP-EPIG       | ACE-US          | ALINE (ours)       |
|--------------|---------------|---------------|---------------|---------------|-----------------|--------------------|
| rpart (6D)   | $0.07 \pm 0.03$ | $0.04 \pm 0.02$ | $0.04 \pm 0.02$ | $0.05 \pm 0.02$ | $\mathbf{0.01} \pm 0.00$ | $\mathbf{0.01} \pm 0.00$ |
| svm (8D)     | $0.22 \pm 0.11$ | $0.11 \pm 0.05$ | $0.12 \pm 0.07$ | $0.15 \pm 0.08$ | $0.04 \pm 0.01$ | $\mathbf{0.03} \pm 0.01$ |
| ranger (9D)  | $0.10 \pm 0.02$ | $0.07 \pm 0.01$ | $0.08 \pm 0.02$ | $0.08 \pm 0.02$ | $\mathbf{0.02} \pm 0.01$ | $\mathbf{0.02} \pm 0.01$ |
| xgboost (16D)| $0.09 \pm 0.02$ | $0.09 \pm 0.02$ | $0.09 \pm 0.02$ | $0.09 \pm 0.02$ | $0.04 \pm 0.01$ | $\mathbf{0.03} \pm 0.01$ |

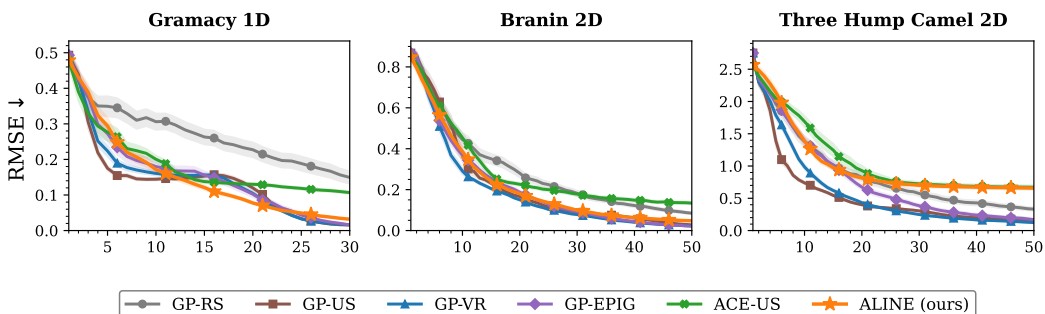

Figure A3: Predictive performance in terms of RMSE on three other active learning benchmark functions. Results show the mean and 95% confidence interval (CI) across 100 runs. Notably, on the Three Hump Camel function, the performance of amortized methods like ALINE and ACE-US is limited, as its output scale significantly differs from the pre-training distribution, highlighting a scenario of distribution shift.

area for future work, such as training ALINE on a broader prior distribution of functions, potentially leading to more universally capable models.

**Acquisition visualization for AL.** To qualitatively understand the behavior of our model, we visualize the query strategy employed by ALINE for AL on a randomly sampled synthetic function Figure A4. This visualization illustrates how ALINE iteratively selects query points to reduce uncertainty and refine its predictive posterior.

**Hyperparameter inference visualization.** We now visualize the evolution of ALINE's estimated posterior distributions for the underlying GP hyperparameters for a randomly drawn 2D synthetic GP function (see Figure A5). The posteriors are shown after 1, 15, and 30 active data acquisition steps. As ALINE strategically queries more informative data points, its posterior beliefs about these generative parameters become increasingly concentrated and accurate.

**Inference time.** To assess the computational efficiency of ALINE, we report the inference times for the AL tasks in Table A4. The times represent the total duration to complete a sequence of 30 steps for 1D functions and 50 steps for 2D functions, averaged over 10 independent runs. As both ALINE and ACE-US perform inference via a single forward pass per step once trained, they are significantly faster compared to traditional GP-based methods.

Table A4: Comparison of inference times (seconds) for different AL methods on 1D (30 steps) and 2D (50 steps) tasks. Values are averaged over 10 runs (mean $\pm$ standard deviation).

| Methods | Inference time (s) | |
|---------|--------------------|----|
|         | 1D & 30 steps | 2D & 50 steps |
| GP-US   | $0.62 \pm 0.09$ | $1.72 \pm 0.23$ |
| GP-VR   | $1.41 \pm 0.14$ | $4.03 \pm 0.18$ |
| GP-EPIG | $1.34 \pm 0.11$ | $3.43 \pm 0.24$ |
| ACE-US  | $0.08 \pm 0.00$ | $0.19 \pm 0.02$ |
| ALINE   | $0.08 \pm 0.00$ | $0.19 \pm 0.02$ |

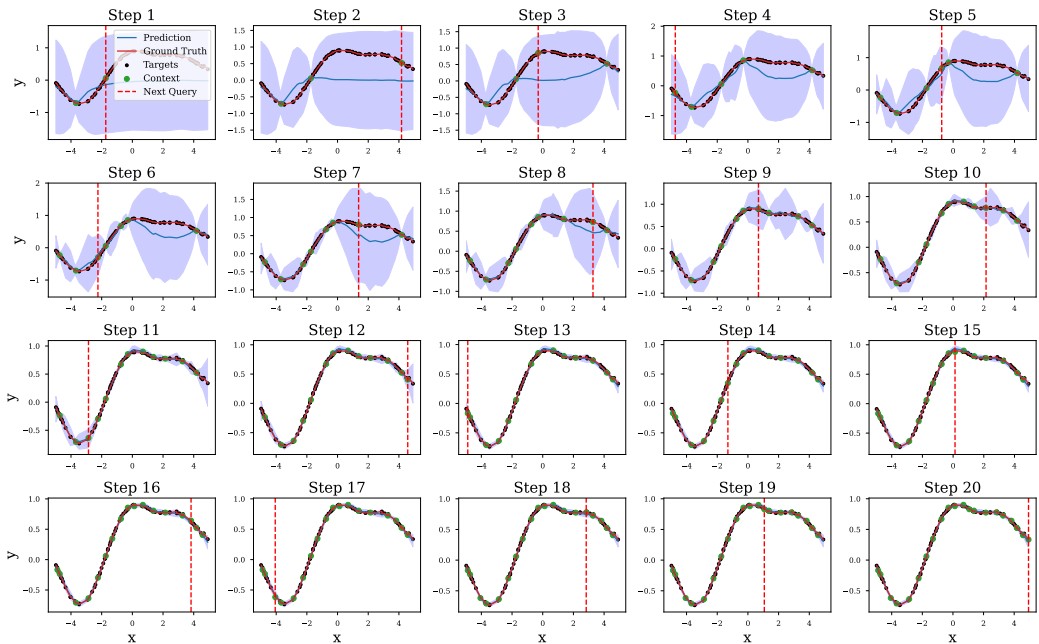

Figure A4: Sequential query strategy of ALINE on a 1D synthetic GP function over 20 steps. As more points are queried, the model's prediction increasingly aligns with the ground truth, and the uncertainty is strategically reduced.

## D.3 Benchmarking on Bayesian experimental design tasks

In this section, we provide additional qualitative results for ALINE's performance on the BED benchmark tasks. Specifically, for the Location Finding task, we visualize the sequence of designs chosen by ALINE and the resulting posterior distribution over the hidden source's location (Figure A6). For the CES task, we present the estimated marginal posterior distributions for the model parameters, comparing them against their true underlying values (Figure A7). We see that ALINE offers accurate parameter inference.

## D.4 Psychometric model

**Demonstrations of flexibility.** We conduct two ablations to explicitly validate ALINE's flexible targeting capabilities.

First, we test the ability to switch targets mid-rollout. We configure a single experiment where for the first 15 steps, the target is *threshold & slope* parameters, and at step 16, the target is switched to the *guess rate & lapse rate*. As shown in Figure A8(a), ALINE's acquisition strategy adapts immediately and correctly, shifting its queries from the decision threshold region to the extremes of the stimulus range to gain maximal information about the new targets.

Second, we test generalization to novel target combinations. A single ALINE model is trained to handle two distinct targets separately: (1) *threshold & slope* and (2) *guess & lapse rate*. At deployment, we task this model with a novel, unseen combination: targeting all four parameters simultaneously. As shown in Figure A8(b), the resulting policy is a sensible mixture of the two underlying strategies it has learned, strategically alternating queries between points near the decision threshold and points at the extremes. This confirms that ALINE can successfully compose its learned strategies to generalize to new inference goals at runtime.

**Inference time.** We additionally assess the computational efficiency of each method in proposing the next design point. The average per-step design proposal time, measured over the 30-step psychometric experiments across 20 runs, is $0.002 \pm 0.00$s for ALINE, $0.07 \pm 0.00$s for QUEST+, and $0.02 \pm 0.00$s for Psi-marginal. Methods like QUEST+ and Psi-marginal, which often rely on

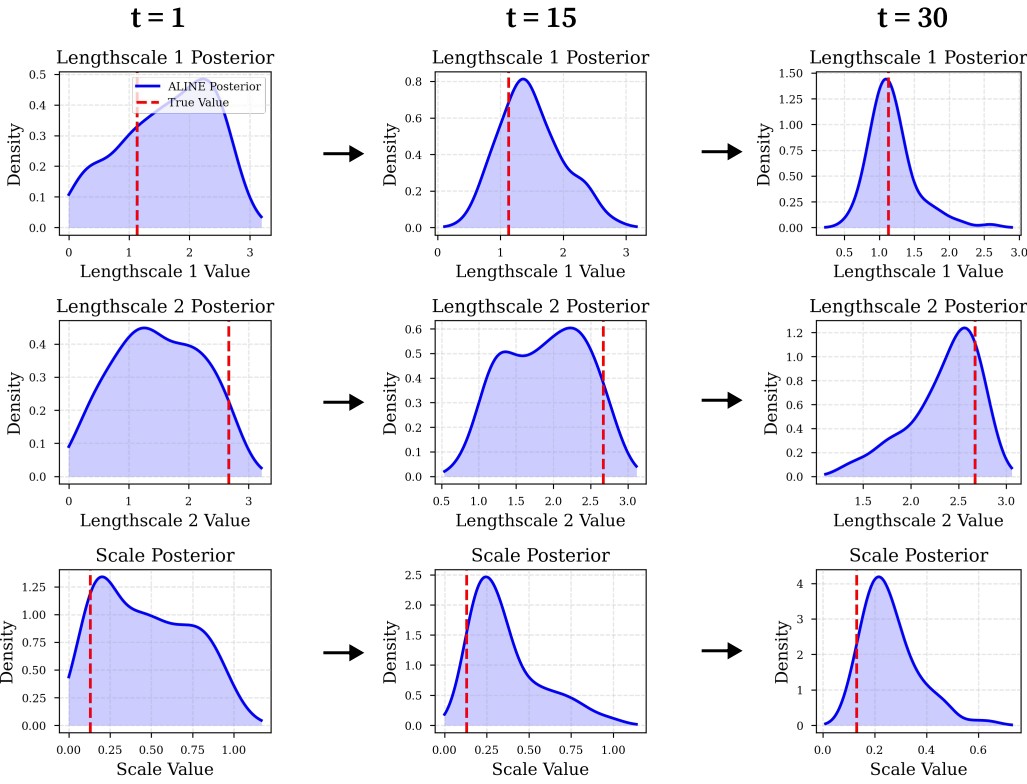

Figure A5: Estimated posteriors for the two lengthscales and the output scale obtained from ALINE after $t = 1$, $t = 15$, and $t = 30$ active query steps. The posteriors progressively concentrate around the true parameter values as more data is acquired.

grid-based posterior estimation, face rapidly increasing computational costs as the parameter space dimensionality or required grid resolution grows. ALINE, however, estimates the posterior via the transformer in a single forward pass, making its inference time largely insensitive to these factors. Thus, this computational efficiency gap is anticipated to become even more pronounced for more complex psychometric models.

## E  Computational resources and software

All experiments presented in this work, encompassing model development, hyperparameter optimization, baseline evaluations, and preliminary analyses, are performed on a GPU cluster equipped with AMD MI250X GPUs. The total computational resources consumed for this research, including all development stages and experimental runs, are estimated to be approximately 5000 GPU hours. For each experiment, it takes around 20 hours to train an ALINE model for $10^5$ epochs. The core code base is built using Pytorch (`https://pytorch.org/`, License: modified BSD license). For the Gaussian Process (GP) based baselines, we utilize Scikit-learn [56] (`https://scikit-learn.org/`, License: modified BSD license). The DAD baseline is adapted from the original authors' publicly available code [23] (`https://github.com/ae-foster/dad`; MIT License). Our implementations of the RL-BOED and vsOED baselines are adapted from the official repositories provided by [6] (`https://github.com/yasirbarlas/RL-BOED`; MIT License) and [65] (`https://github.com/wgshen/vsOED`; MIT License), respectively. We use `questplus` package (`https://github.com/hoechenberger/questplus`, License: GPL-3.0) to implement QUEST+, and use `Psi-staircase` (`https://github.com/NNiehof/Psi-staircase`, License: GPL-3.0) to implement the Psi-marginal method.

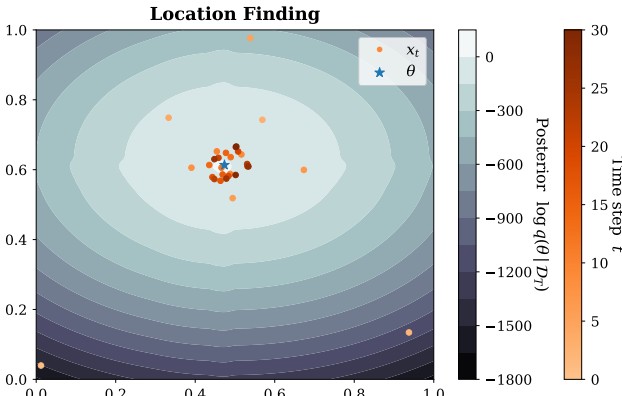

Figure A6: Visualization of ALINE's design policy and resulting posterior for the Location Finding task. The contour plot shows the log posterior probability density of the source location $\theta$ (true location marked by blue star) after $T = 30$ steps. Queried locations, with color indicating the time step of acquisition, demonstrating a concentration of queries around the true source.

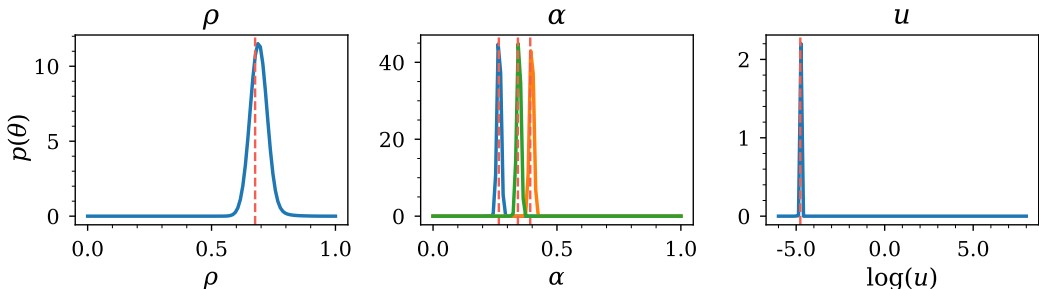

Figure A7: ALINE's estimated marginal posterior distributions for the parameters of the CES task after $T = 10$ query steps. The dashed red lines indicate the true parameter values. The posteriors are well-concentrated around the true values, demonstrating accurate parameter inference.

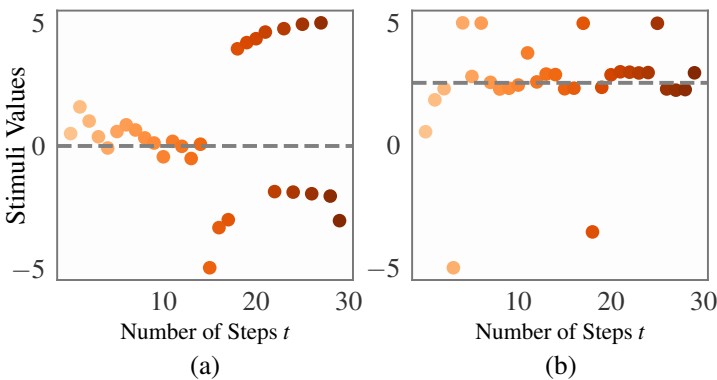

Figure A8: Demonstration of ALINE's runtime flexibility on the psychometric task. (a) The acquisition strategy adapts after the inference target is switched mid-rollout from (threshold & slope) to (guess & lapse rate). (b) When tasked with a novel combined target (all four parameters), the policy generalizes by mixing the two distinct strategies it learned during training.

