# OpenReview forum: "ALINE: Joint Amortization for Bayesian Inference and Active Data Acquisition"
_NeurIPS.cc/2025/Conference — NeurIPS 2025 spotlight_

### Official Review · Reviewer_GfQV · 2025-07-01

**Clarity:** 4
**Significance:** 3
**Originality:** 3
**Rating:** 5
**Confidence:** 4

**Summary:**

This paper develops the Amortized Active Learning and INference Engine (ALINE) which employs a sim2real training setup to train a transformer-based backbone to simultaneously perform amortised Bayesian inference and amortised active data acquisition. Though a flexible approach to conditioning the model can be deployed on a range of different acquisition tasks. ALINE is compared to standard GP approaches on a GP regression / OOD task, it is evaluated on two classical Bayesian experimental design tasks, and a psychometric experimental design task.

**Questions:**

The RL component of the objective depends on the model predictions e.g. q(\theta | D_c), but are gradients of the RL loss wrt q used in training or are these stopped? It seems like you only want policy gradients flowing through this objective?

How well does the approach generalise to scenarios (e.g. combinations of targets) at deployment time?

**Ethical Concerns:**

["NO or VERY MINOR ethics concerns only"]

**Final Justification:**

I think this is a solid paper and I vote for acceptance.

**Limitations:**

Yes

**Quality:**

3

**Strengths And Weaknesses:**

Strengths

The paper is tightly and clearly written and well-polished. I like table 1 for situating the current work in the context of past work. The method was neat with a single transformer-based backbone to perform three distinct tasks (forming the predictive, inferring parameters, and implementing the policy). The experiments were also well done and I especially liked the final experiment on experimental design for efficiently eliciting psychometric functions.

Weaknesses

Whereas neural processes can be trained directly on real data without the need for sim2real transfer, the current approach to supporting inference for parameters generally relies on a sim2real setup as it requires access to ground truth parameters at training time. This is not necessarily a huge limitation (we have to specify models when performing Bayesian inference and the TabPFN line have work has shown the sim2real transfer in NPs can work well for the predictive), but it does nevertheless require the sim2real gap to be bridged (which regular Bayesian inference does not have to address). I’d point this out more centrally in the development of the paper. At the moment this is mentioned a little cryptically in the “Limitations section” whereas I think it should be stated up front as a key aspect of the method.

I terms of novelty, I thought that although the method was a combination of well-known components, the combination was quite sophisticated and sensibly put together.

Minor point: In the related work I’d mention "Environmental sensor placement with convolutional Gaussian neural processes” which uses Gaussian neural processes for active learning to support sensor placement and performs well compared to standard baselines.

https://www.cambridge.org/core/journals/environmental-data-science/article/environmental-sensor-placement-with-convolutional-gaussian-neural-processes/F466DBCE3FA1088E04335D98225FA572

Figure 2 / Section 5.1: It was good to see the comparison to the more standard approach of using a GP with an acquisition function, but I was a bit surprised that there was not a comparison to an existing neural process based active learning approach. This might enable you to decouple improvements that arise because of the use of the neural process like architecture and improvements that arise because of the more sophisticated policy used for acquisition.

e.g. see section 5 in "Meta-Learning Stationary Stochastic Process
Prediction with Convolutional Neural Processes” or the reference mentioned above. https://papers.nips.cc/paper/2020/file/5df0385cba256a135be596dbe28fa7aa-Paper.pdf

More generally, although I liked the experiments that were present in the paper, I thought the experiments were slightly on the light side.

---

> ### Author Rebuttal · Authors · 2025-07-30
>
> We sincerely thank you for your positive assessment of our paper. We appreciate your thoughtful suggestions and address your questions below.
>
> > *1. Whereas neural processes can be trained directly on real data without the need for sim2real transfer, the current approach to supporting inference for parameters generally relies on a sim2real setup as it requires access to ground truth parameters at training time. ... I’d point this out more centrally in the development of the paper. At the moment this is mentioned a little cryptically in the “Limitations section” whereas I think it should be stated up front as a key aspect of the method.*
>
> Excellent point. We are indeed implicitly assuming that the model we are generating training data from is “well-specified”, which can be violated, thus requiring the sim2real gap to be bridged. We will state this assumption more prominently at the beginning of the method section.
>
> > *2. In the related work I’d mention "Environmental sensor placement with convolutional Gaussian neural processes” which uses Gaussian neural processes for active learning to support sensor placement and performs well compared to standard baselines.*
>
> Thank you for pointing out this reference.  The connection to our work is that both approaches leverage the power of Neural Processes to move beyond the limitations of traditional GPs in active learning. And we believe this interesting real-world application could provide a compelling potential use case for ALINE. We will gladly add this work to our related work section.
>
> > *3.  Figure 2 / Section 5.1: It was good to see the comparison to the more standard approach of using a GP with an acquisition function, but I was a bit surprised that there was not a comparison to an existing neural process based active learning approach. This might enable you to decouple improvements that arise because of the use of the neural process like architecture and improvements that arise because of the more sophisticated policy used for acquisition.*
>
> We fully agree. In fact, we did include such a baseline (ACE-US) in our active learning experiments. ACE (Amortized Conditioning Engine) [1] is a very recent, powerful **transformer-based neural process** that extends earlier work [2, 3]. We specifically chose it as the foundation for our baseline because, unlike many earlier NPs, it can perform amortized inference on both data (posterior predictive) and latent parameters (posterior), making it a fair and direct architectural counterpart to ALINE. We recognize and apologize that we did not make the role and nature of the ACE-US baseline clearer in our submission. We will revise the experimental section to explicitly mention ACE as an amortized neural process model and emphasize that this baseline serves this important ablative purpose.
>
> > *4. More generally, although I liked the experiments that were present in the paper, I thought the experiments were slightly on the light side.*
>
> Thank you for the comment. To strengthen our evaluation, we have conducted a new set of experiments on a practical high-dimensional task: **Active Exploration of Hyperparameter Performance Landscapes**. The task is to actively query a small number of hyperparameter configurations to build a surrogate model that accurately predicts performance for a larger, held-out set of target configurations.
>
> We used the high-dimensional, real-world tasks from the **HPO-B meta benchmark** [4]. Specifically, we evaluated on the rpart (6D), svm (8D), ranger (9D), and xgboost (16D) datasets. ALINE was trained on their multiple pre-defined training sets. We then evaluated its performance, alongside non-amortized GP-based baselines and an amortized surrogate baseline (ACE-US), on the benchmark's held-out and entirely unseen test sets. This setup is explicitly designed to test the model's ability to generalize to new problems.
>
> The table below shows the RMSE with 95% confidence interval after 30 steps, averaged across all test tasks for each dataset. First, **both amortized methods, ALINE and ACE-US, significantly outperform all non-amortized GP-based baselines** across all tasks. This highlights the power of meta-learning in this domain. GP-based methods must learn each new performance landscape from scratch, which is highly inefficient in high dimensions. In contrast, both ALINE and ACE-US are pre-trained on hundreds of related tasks and their Transformer architectures meta-learn the structural patterns common to these landscapes. This shared prior knowledge allows them to make far more accurate predictions from sparse data. Second, while ACE-US performs strongly due to its amortized nature, **ALINE consistently achieves the best or joint-best performance.** This demonstrates the additional, crucial benefit of our core contribution: the learned acquisition policy. ACE-US relies on a standard heuristic, whereas ALINE's policy is trained end-to-end to learn how to optimally explore the landscape, leading to more informative queries and ultimately a more accurate final surrogate model.
>
> We believe this new experiment, alongside the existing BED and psychometric tasks, now provides a comprehensive and compelling demonstration of ALINE's practical utility. We will add these results to the revised manuscript.
>
> | RMSE ($\\downarrow$) | GP-RS | GP-US | GP-VR | GP-EPIG | ACE-US | ALINE |
> | :---- | :---- | :---- | :---- | :---- | :---- | :---- |
> | rpart (6D) | 0.07±0.03 | 0.04±0.02 | 0.04±0.02 | 0.05±0.02 | **0.01±0.00** | **0.01±0.00** |
> | svm (8D) | 0.22±0.11 | 0.11±0.05 | 0.12±0.07 | 0.15±0.08 | 0.04±0.01 | **0.03±0.01** |
> | ranger (9D) | 0.10±0.02 | 0.07±0.01 | 0.08±0.02 | 0.08±0.02 | **0.02±0.01** | **0.02±0.01** |
> | xgboost (16D) | 0.09±0.02 | 0.09±0.02 | 0.09±0.02 | 0.09±0.02 | 0.04±0.01 | **0.03±0.01** |
>
> > *5. The RL component of the objective depends on the model predictions e.g. $q(\theta | D_c)$, but are gradients of the RL loss wrt q used in training or are these stopped? It seems like you only want policy gradients flowing through this objective?*
>
> When updating the acquisition policy, we **do stop the gradients** from the RL loss with respect to the parameters of the inference network. This is a deliberate implementation choice for the REINFORCE algorithm in this context, and it is crucial for stable training. The rationale is to ensure each component has a clear and distinct objective:
>
> * The **policy gradient** updates only the parameters of the **acquisition policy**, encouraging it to select actions that lead to higher rewards.
> * The parameters of the **inference network**, in turn, are updated *only* by the **negative log-likelihood loss**, which has the sole objective of pushing to become a more accurate approximation of the target distribution.
>
> Allowing gradients to flow from the policy loss back into the inference network would create a problematic objective. The policy would learn not only to find informative points but also to directly manipulate the inference network to report higher rewards, potentially distorting the posterior estimates. We will add a sentence to the manuscript to explicitly clarify this important implementation detail.
>
> > *6. How well does the approach generalise to scenarios (e.g. combinations of targets) at deployment time?*
>
> Theoretically, ALINE is designed to generalize to novel target combinations at deployment time, even if those specific combinations were not seen during training. This is a direct benefit of our **query-target cross-attention mechanism**. It allows each candidate query point to compute its relevance and potential information gain with respect to each target embedding individually and in parallel. The final policy is then based on an aggregation of these individual relevance scores.
>
> To empirically validate this, we ran a new experiment on the psychometric task.
>
> * **Training:** We trained a single ALINE model to handle two distinct targets separately: (1) the "threshold & slope" and (2) the "guess & lapse rate".
> * **Testing:** At deployment, we tasked this model with a novel, unseen combination: to target **all four parameters simultaneously**.
>
> We observed that the resulting acquisition policy was a sensible mixture of the two underlying strategies it had learned. It strategically alternated its queries between points near the decision threshold (optimal for the first target) and points at the extremes (optimal for the second). This behavior confirms that ALINE can successfully compose its learned strategies to generalize to new combinations of inference goals at runtime. We will include this experiment in the revised manuscript.
>
> **References**
>
> [1] Chang, et al., Amortized Probabilistic Conditioning for Optimization, Simulation and Inference, AISTATS (2025).
>
> [2] Müller, et al., Transformers Can Do Bayesian Inference, ICLR (2022).
>
> [3] Nguyen and Aditya, Transformer neural processes: Uncertainty-aware meta learning via sequence modeling, ICML (2022).
>
> [4] Arango, et al., HPO-B: A Large-Scale Reproducible Benchmark for Black-Box HPO based on OpenML, NeurIPS Track on Datasets and Benchmarks (2021).

---

> > ### Comment · Reviewer_GfQV · 2025-08-04
> > **Thanks!**
> >
> > Thanks for the clarifications and the extra experiments.
> >
> > I'm very happy to maintain my strong score.

---

> > > ### Author Response · Authors · 2025-08-04
> > >
> > > Thank you for the positive feedback. We are very grateful for your strong support of our work.

---

### Official Review · Reviewer_xxdt · 2025-07-02

**Clarity:** 3
**Significance:** 2
**Originality:** 3
**Rating:** 5
**Confidence:** 3

**Summary:**

Amortization is a powerful tool in Bayesian inference and more recently in Bayesian experimental design. The paper proposes a unified approach for amortized Bayesian inference and data acquisition and is designed to handle both predictive and parameter targets. Specifically, the proposed method Aline, given an accumulated history and a target (predictive or parameter) can pick the next data point to acquire and estimate the posterior (or posterior predictive). To achieve this, Aline uses a modification of the Transformer Neural Processes architecture, with separate heads for the acquisition policy and inference. The method uses the sEIG bound from DAD, and a variant of EPIG called sEPIG for the predictive case as the objectives for data acquisition policy and a factorized likelihood for the inference head. The training of the acquisition policy is framed as a RL problem to maximize the variational lower bounds (which define the per step rewards). The method is evaluated on some synthetic benchmarks for active learning and standard tasks used in the Bayesian experimental design literature. Aline outperforms the task specific baselines in most experiments. The authors finally present results on a psychometric model with domain specific baselines and show that Aline matches the performance of these baseline while being computationally more efficient.

**Questions:**

* The experiments only consider problems with continuous $x$ and $y$ and it is unclear how the method applies to cases where $x$ is discrete for instance?
* Is it possible to switch between predictive and parameter targets in the middle of a rollout?
* Could you elaborate on potential ways to make the method support inputs with arbitrary dimensionality?
* What are the most important bottlenecks for scaling this to more complicated and higher dimensional problems?

**Ethical Concerns:**

["NO or VERY MINOR ethics concerns only"]

**Final Justification:**

The reviewers answered a lot of the questions I had and ran several experiments on realistic and slightly higher dimensional tasks. The fundamental limitations in the current setup remain but the paper is stronger so I have increased my score.

**Limitations:**

Yes

**Quality:**

3

**Strengths And Weaknesses:**

Strengths:
* To the best of my knowledge, this work is the first to propose joint training of an amortized acquisition and inference network. I really like this idea since it unifies these two very intimately linked procedures and demonstrates the benefits of exploiting the structure between the two.
* Another strength of the approach is that in addition to allowing specification of predictive or parameter targets, it also allows specification of subsets of parameters as the target which can provide a lot of flexibility in problems with known parameter structure.
* The paper was extremely well written. The problem is clearly set up and each component of the approach is described clearly in sufficient detail.
* On the tasks considered, the performance is similar to or slightly better than task specific methods while being faster in terms of the training time than some of the baselines and competitive in terms of the deployment time.
* The authors also include the code which is helpful for reproducibility, in. addition to very detailed description in the paper itself.

Weaknesses:
* One of the main shortcomings of the paper is limited the scale of the empirical evaluations. All the experiments are limited to fairly low dimensional problems. I do understand these amortized networks can be expensive to train but I believe at demonstrating at least some scalability can be very useful.
* Another weakness which is already mentioned in the paper is the choice of assuming factorized likelihood over the targets. While a potential solution in the form of an autoregressive extenions is discussed, it is left for future work.
* While the training procedures for the individual parts is sound and well justified, there is not discussion of the stability of the joint training procedure. Based on my understanding there is no guarantee for the joint training procedure to converge (though I am a bit unsure about this and happy to hear the authors' thoughts).

---

> ### Author Rebuttal · Authors · 2025-07-30
>
> Thank you for your positive feedback and questions. We address each of them in detail below.
>
> > *1. One of the main shortcomings of the paper is the limited scale of the empirical evaluations.*
>
> To address the scalability concern, we have conducted a new set of experiments on a high-dimensional task: **Active Exploration of Hyperparameter Performance Landscapes.** The task is to actively query a small number of hyperparameter configurations to build a surrogate model that accurately predicts performance for a set of target configurations.
>
> We use the real-world tasks from the HPO-B meta-benchmark [1]. Specifically, we evaluate on the *rpart* (6D), *svm* (8D), *ranger* (9D), and *xgboost* (16D) datasets. ALINE is trained on their multiple pre-defined training sets. We then evaluate it alongside non-amortized GP-based baselines and an amortized surrogate baseline (ACE-US), on the benchmark's entirely unseen test sets.
>
> The table below shows the RMSE with 95% CI after 30 steps, averaged across all test sets. First, **both amortized methods significantly outperform all GP-based baselines** across all tasks. This highlights the power of meta-learning in this domain. GP-based methods must learn each new performance landscape from scratch, which is highly inefficient in high dimensions. In contrast, both ALINE and ACE-US are pre-trained on hundreds of related tasks and meta-learn the structural patterns common to these landscapes. This shared prior knowledge allows them to make more accurate predictions from sparse data. Second, while ACE-US performs strongly due to its amortized nature, **ALINE consistently achieves the best or joint-best performance.** This demonstrates the additional benefit of our core contribution: the learned acquisition policy. ALINE's policy is trained end-to-end to learn how to optimally explore the landscape, leading to more informative queries and ultimately a more accurate final surrogate model.
>
> We believe this new experiment provides a compelling demonstration of ALINE's scalability. We will add a full description of these results to the revised manuscript.
>
> | RMSE ($\\downarrow$) | GP-RS | GP-US | GP-VR | GP-EPIG | ACE-US | ALINE |
> | :---- | :---- | :---- | :---- | :---- | :---- | :---- |
> | rpart (6D) | 0.07±0.03 | 0.04±0.02 | 0.04±0.02 | 0.05±0.02 | **0.01±0.00** | **0.01±0.00** |
> | svm (8D) | 0.22±0.11 | 0.11±0.05 | 0.12±0.07 | 0.15±0.08 | 0.04±0.01 | **0.03±0.01** |
> | ranger (9D) | 0.10±0.02 | 0.07±0.01 | 0.08±0.02 | 0.08±0.02 | **0.02±0.01** | **0.02±0.01** |
> | xgboost (16D) | 0.09±0.02 | 0.09±0.02 | 0.09±0.02 | 0.09±0.02 | 0.04±0.01 | **0.03±0.01** |
>
> > *2. Another weakness which is already mentioned in the paper is the choice of assuming factorized likelihood over the targets.*
>
> You are correct, and we acknowledge this choice as a limitation in the paper. Our decision to adopt a factorized approximation was a deliberate and critical **trade-off between modeling accuracy and computational tractability**.
>
> An autoregressive model would require sequential generation of each marginal dimension of the posterior, which is computationally prohibitive within our RL framework. Calculating the reward for $M$ targets would require $M$ **sequential forward passes** through the Transformer, making the training process **infeasibly slow**. Therefore, we adopted the factorized approximation - a common and efficient practice in the Neural Process literature [2, 3, 4] - which allows for computation in a single parallel forward pass. Our strong empirical results validate this trade-off between modeling accuracy and computational tractability. We will add a summary of this rationale to the revised manuscript.
>
> > *3. There is no discussion of the stability of the joint training procedure.*
>
> Our approach to ensuring stability is grounded in both the complementary nature of our learning tasks and a key practical implementation detail:
>
> * We can view the joint training as a multi-task learning problem where the two tasks are highly complementary, not conflicting. They mutually regularize the shared Transformer. A better inference model provides a more reliable reward signal for the policy, and a better policy gathers more informative data for the inference model. This synergistic relationship naturally promotes stable learning. We also find this observation in Bayesian optimization literature [5, 6].
> * From a practical standpoint, we found the model to be remarkably straightforward to train. A crucial trick for this is the **initial warm-up phase** where we first train only the inference network. By doing so, we ensure that the inference head provides a stable reward signal before we begin training the acquisition policy. This stable reward landscape is also why we were able to successfully use the basic REINFORCE algorithm, obviating the need for more complex RL methods.
>
> We believe our strong empirical results provide substantial evidence for the method's viability. We agree that a deeper theoretical study would be a valuable direction for future work.
>
> > *4. The experiments only consider problems with continuous x and y and it is unclear how the method applies to cases where x is discrete for instance?*
>
> ALINE can be straightforwardly adapted to handle problems with **discrete inputs and outputs** with minor, standard modifications to the input embedding and output head layers, respectively. Recent Transformer-based neural processes, such as ACE [4], already demonstrate this capability.
>
> * **For discrete inputs:** Instead of using an MLP embedder, we could employ a learnable embedding layer. Each discrete value in the input space would be mapped to a unique, trainable embedding vector. These embedding vectors would then be processed by the Transformer backbone.
> * **For discrete outputs:** The modification would be in the inference head. Instead of parameterizing a GMM, the head would be adapted to output logits corresponding to each possible discrete value. A **Softmax function** would then be applied to these logits to produce a categorical probability distribution over the discrete outcomes.
>
> > *5. Is it possible to switch between predictive and parameter targets in the middle of a rollout?*
>
> Yes, absolutely. This is a key feature of ALINE's flexible design. To switch targets, the user simply has to change this input specifier at the desired step.
>
> To explicitly demonstrate this capability, we have run a new experiment in the psychometric task. We configured a rollout where, for the first 5 steps, the target was the **threshold & slope**, and at step 6, we switched the target to the **guess rate & lapse rate**. We present the sequence of queries in the table below as per the rebuttal format's limitations. The threshold value is 0.49 in this experiment. We observed that ALINE's acquisition strategy adapted immediately from querying the extremes to the values concentrated near the threshold. We will add this new experiment with full 30 steps and its plots to the revised manuscript.
>
> | Step | 1 | 2 | 3 | 4 | 5 | 6 | 7 | 8 | 9 | 10 |
> | :---- | :---- | :---- | :---- | :---- | :---- | :---- | :---- | :---- | :---- | :---- |
> | Queried Stimulus Value | \-3.93 | \-4.92 | 4.70 | 3.04 | 3.06 | \-0.31 | 0.86 | 1.60 | 1.03 | 0.3 |
>
> > *6. Could you elaborate on potential ways to make the method support inputs with arbitrary dimensionality?*
>
> There is already some recent work that can support arbitrary dimensions. A standard approach would be to adopt a **bi-dimensional Transformer** [7, 8], the process would involve a **feature-wise Transformer** that operates across the different feature dimensions of each data point. Then we can use a **point-wise Transformer**, which operates across the sequence of data points as it does now.
>
> This double-Transformer approach has been shown to be quite effective. Integrating this into ALINE is a very feasible direction for future research.
>
> > *7. What are the most important bottlenecks for scaling this to more complicated and higher dimensional problems?*
>
> We see the most significant bottlenecks falling into three main categories:
>
> * **Architectural Flexibility and Scale:** Tackling more complex problems will necessitate significantly larger networks than the relatively small model used here.
> * **Quality of the Training "Meta-Prior":** The performance of ALINE is fundamentally dependent on the diversity and representativeness of the simulated problems used for training. A key bottleneck is the generation of a sufficiently rich training distribution that can bridge the "sim2real" gap and ensure robust generalization to real-world tasks that may differ from the training set.
> * **Computational and Algorithmic Scalability:** Scaling up the model, data, and task complexity will demand substantial computational resources. Algorithmically, moving beyond tractable assumptions (like the factorized likelihood) to more principled but costly methods (like fully autoregressive models) will require significant advances in training efficiency to be feasible at a large scale.
>
> In summary, we believe overcoming these challenges is a major focus for the entire field and a promising direction for future research.
>
> **References**
>
> [1] Arango, et al., HPO-B: A Large-Scale Reproducible Benchmark for Black-Box HPO based on OpenML, NeurIPS Track on Datasets and Benchmarks (2021).
>
> [2] Garnelo, et al., Conditional neural processes, ICML (2018).
>
> [3] Müller, et al., Transformers Can Do Bayesian Inference, ICLR (2022).
>
> [4] Chang, et al., Amortized Probabilistic Conditioning for Optimization, Simulation and Inference, AISTATS (2025).
>
> [5] Maraval, et al., End-to-end meta-bayesian optimisation with transformer neural processes, NeurIPS (2023).
>
> [6] Zhang, et al., PABBO: Preferential Amortized Black-Box Optimization, ICLR (2025).
>
> [7] Lee, et al., Dimension agnostic neural processes, ICLR (2025).
>
> [8] Qu, et al., Tabicl: A tabular foundation model for in-context learning on large data, ICML (2025).

---

> > ### Comment · Reviewer_xxdt · 2025-08-02
> >
> > Thanks for the detailed response! The rebuttal answers most of my questions satisfactorily!
> >
> > I appreciate the additional experiment on HPO-B, but have a few questions to better understand the experiment.
> > * What exactly is the setup you are considering - is it the HPO task?
> > * If it is the HPO task, why are there no standard BO baselines like GP-UCB?
> > * It is not clear what the training strategy here is - do you train ALINE on each of these tasks separately (since it only supports fixed dimensionality for now)?

---

> > > ### Author Response · Authors · 2025-08-02
> > >
> > > Thank you for the follow-up questions about our new experiment. We are happy to provide these clarifications.
> > >
> > > > *1. What exactly is the setup you are considering - is it the HPO task?*
> > >
> > > The setup we consider is an **Active Learning (AL) task for pure exploration**, consistent with Section 5.1 of our paper, and not a Bayesian Optimization (BO) task. Please note that the objective is *not* to find the single best hyperparameter configuration. Instead, the goal is to **build the most accurate surrogate model of the performance landscape** with a limited query budget, in order to accurately predict the performance of a held-out set of target configurations.
> > >
> > > While the HPO-B benchmark was originally designed for hyperparameter optimization, we selected it for evaluating our active learning approach because HPO and AL share fundamental characteristics: both involve strategic exploration of high-dimensional spaces to efficiently identify regions of interest. HPO-B provides exactly what challenging AL evaluation requires - numerous diverse performance landscapes with complex, multi-modal structures that challenge exploration strategies.
> > >
> > > Since our focus is on the exploration aspect of these landscapes rather than finding optimal configurations, we compared against AL acquisition functions (such as GP-US, GP-EPIG) rather than optimization-focused methods. These baselines better reflect our objective of efficiently learning about the entire performance surface rather than quickly converging to optima.
> > >
> > > > *2. If it is the HPO task, why are there no standard BO baselines like GP-UCB?*
> > >
> > > Indeed, as mentioned above, this is not the standard HPO task, but an AL variant of it. We will clarify further in the paper that standard BO acquisition functions like GP-UCB are designed to balance exploration and exploitation to find optima efficiently. However, here our objective is pure exploration - learning the entire performance surface rather than locating its peaks. The exploitation component of UCB, which biases sampling toward promising regions, actually works against this goal by reducing coverage of the full landscape. This fundamental mismatch makes UCB a less appropriate baseline than pure exploration methods like GP-US and GP-EPIG, which are specifically designed for uncertainty reduction and information gain across the entire space.
> > >
> > > That said, the empirical performance of GP-UCB on pure exploration tasks remains an interesting open question worth investigating. Motivated by your question, **we conducted additional experiments with GP-UCB** using the same $\beta$ value as in the original HPO-B paper.
> > >
> > > The table below shows the results alongside GP-US (our strongest exploration-based GP baseline) and ALINE. Indeed, GP-UCB underperforms both exploration-focused methods on this predictive task, confirming that the exploitation-exploration balance optimized for finding optima does not transfer well to learning entire landscapes. ALINE achieves the best performance overall, demonstrating the value of our approach.
> > >
> > > | RMSE ($\\downarrow$) | GP-UCB | GP-US | ALINE |
> > > | :---- | :---- | :---- | :---- |
> > > | rpart (6D) | 0.09±0.03 | 0.04±0.02 | **0.01±0.00** |
> > > | svm (8D) | 0.15±0.07 | 0.11±0.05 | **0.03±0.01** |
> > > | ranger (9D) | 0.13±0.04 | 0.07±0.01 | **0.02±0.01** |
> > > | xgboost (16D) | 0.12±0.04 | 0.09±0.02 | **0.03±0.01** |
> > >
> > > > *3. Do you train ALINE on each of these tasks separately?*
> > >
> > > Yes. Since the current version of ALINE is not dimension-agnostic, **we trained a separate ALINE model for each meta-dataset.** We believe extending ALINE to be dimension-agnostic is an interesting direction for future work.
> > >
> > > We hope this clarifies the setup for our new experiment. We will ensure these details are explained clearly in the revised manuscript. Thank you again for your insightful questions.

---

> > > > ### Comment · Reviewer_xxdt · 2025-08-02
> > > >
> > > > Thanks for the clarification. I was considering it to be the HPO task and hence my confusion. In light of the rebuttal I will increase my score.

---

> > > > > ### Author Response · Authors · 2025-08-02
> > > > >
> > > > > Thank you for engaging with our rebuttal and raising your score. We greatly appreciate it.

---

### Official Review · Reviewer_gxPy · 2025-07-03

**Clarity:** 3
**Significance:** 2
**Originality:** 2
**Rating:** 4
**Confidence:** 4

**Summary:**

This paper proposes ALINE, amortized active learning and inference engine, a unified framework that jointly performing amortized Bayesian inference and optimal experimental design. ALINE employs a transformer-based architecture with two specialized heads, an inference head for variational posterior inference and an acquisition head for data selection policy. The authors leverage a query-target cross-attention mechanism that enables flexible targeting of specific parameter subsets or prediction goals, rather than optimize the expected information gain (EIG) over all the parameters. ALINE is trained using reinforcement learning (RL) where the reward is to tighten the variational posterior lower bound.

**Questions:**

Please see the comments in the Weaknesses part.

**Ethical Concerns:**

["NO or VERY MINOR ethics concerns only"]

**Final Justification:**

The authors responded to my concern by discussing the relevant related work and adding new comparison experiments. My main concern has been addressed, and I have accordingly increased my score.

**Limitations:**

The authors discussed the limitations in the Section 6. They did not mention the Broader impacts but provided justification in the Checklist.

**Paper Formatting Concerns:**

I do not find any formatting issues.

**Quality:**

2

**Strengths And Weaknesses:**

__Strengths__:

1. ALINE presents a unified framework to jointly amortize both posterior inference and Bayesian optimal experimental design (BOED) within a single neural architecture. The dual-head architecture (inference head for posterior estimation and acquisition head for design policy) with joint training is well-motivated and more align with the practical setting, where the ultimate goal of performing BOED is to get an informative posterior for inference under a limited budget.

2. The query-target cross-attention mechanism enables ALINE to dynamically adapt its acquisition strategy at runtime to focus on specific parameter subsets or predictive goals..

__Weaknesses__:

1. This paper has significant overlap with the already-published vsOED work [1]. Both papers: (a) tackle sequential BOED with RL-based amortization, (b) handle flexible design criteria and prediction goals, (c) use similar reward formulations (ALINE's incremental reward is essentially equivalent to vsOED's variational-one-point-IIG), (d) evaluate on identical benchmarks (Location Finding) against the same baselines (DAD, RL-BOED), and (e) claim similar efficiency advantages. From my perspective, the contribution of ALINE is the unified architecture for both posterior inference and BOED and the query-target cross-attention mechanism applied in the framework. However, the core problem motivation and methodological approach are no longer novel given [1]'s prior publication. To properly claim the contribution of this paper, the authors should add a thorough discussion comparing the ALINE and vsOED, and also include vsOED in the baselines for a direct empirical comparison.

2. ALINE requires explicit likelihood for training (due to log-probability computations in the reward signal and training loss), while vsOED [1] handles both explicit and implicit likelihood cases through its actor-critic framework. This makes ALINE more limited than existing work.

3. The paper lacks formal theoretical analysis regarding convergence properties, optimality guarantees, or theoretical relationships to information-theoretic objectives.

4. (typo) Line 210: Should the "locations" here be deleted as not specifically discussing the Location Finding task here?


[1] Shen, W., Dong, J., & Huan, X. (2025). Variational sequential optimal experimental design using reinforcement learning. Computer Methods in Applied Mechanics and Engineering, 444, 118068.

---

> ### Author Rebuttal · Authors · 2025-07-30
>
> Thank you for your constructive feedback. Below, we respond to the specific points raised.
>
> > *1. This paper has significant overlap with the already-published vsOED work.*
>
> Thank you for pointing out this relevant vsOED [1] paper. There indeed are high-level similarities, particularly in using an RL framework with variational rewards for sequential design (similar to the RL-sCEE [2] we mentioned in our paper).
>
> However, we are confident that ALINE's core contributions are distinct and complementary to those of vsOED. To provide a direct and concrete comparison, **we have now added vsOED as a new baseline in our BED experiments.** Below, we outline the key distinctions and present the outcome of our new experiments.
>
> * The most critical distinction is ALINE's **dynamic targeting capability at runtime**. A single, pre-trained ALINE policy can instantly adapt its strategy to *any* user-specified target simply by changing an input to the model. vsOED, while flexible in its problem formulation (e.g., weighting between model and parameter EIG with its $\\alpha$ coefficients), would require training a new policy for each different design criterion. This on-the-fly adaptability is a central contribution of our work.
> * ALINE introduces **a unified architecture** for both the inference and acquisition tasks. This design is not only computationally efficient by sharing a single Transformer backbone, but we also believe it benefits from jointly learning the two tasks (as elaborated in our response to Question #2 of Reviewer 9JtZ). The complementary objectives act as a form of mutual regularization, encouraging the model to learn a richer data representation. We believe this contributes to ALINE's strong performance, which is supported by our new empirical results showing that **ALINE outperforms vsOED on shared BED tasks**.
> * While vsOED focuses on BED tasks, ALINE's unified framework naturally extends to active learning for predictive tasks. We introduce the **sequential Expected Predictive Information Gain (sEPIG)** and a variational lower bound for it. This is a specific methodological contribution aimed at the active learning domain.
>
> **New Empirical Comparison:** In our new experiments on the Location Finding and CES tasks, ALINE consistently achieves a higher PCE than vsOED (see the table below). For transparency, our re-implementation of vsOED achieves performance consistent with the results reported in their paper, which validates our comparison.
>
> | PCE ($\\uparrow$) | vsOED | ALINE |
> | :---- | :---- | :---- |
> | Location Finding | 7.30±0.06 | **8.91±0.04** |
> | CES | 12.12±0.18 | **14.37±0.08** |
>
> Thank you again for prompting this important comparison. In our revised manuscript, we will add a summary of the above discussion of vsOED in the introduction and related work, and include it as a baseline in our results to make these distinctions clear.
>
> > *2. ALINE requires explicit likelihood for training (due to log-probability computations in the reward signal and training loss), while vsOED [1] handles both explicit and implicit likelihood cases through its actor-critic framework. This makes ALINE more limited than existing work.*
>
> We would like to respectfully clarify that this is a misunderstanding. ALINE **does not require an explicit or differentiable likelihood and naturally supports implicit likelihood models**, a key feature it shares with vsOED.
>
> Our reward signal, defined in **Eq. 10**, is derived from the log-probability of the ground truth parameters from the simulation, under our **variational posterior approximation** predicted by the neural network, i.e., $\log q(\theta|D)$. During training, we **never require evaluating the model's likelihood $p(y|\theta,d)$**. The only interaction with the true generative process is the ability to **sample outcomes** ($y \sim p(y|\theta, x)$) to construct the experimental history. This is a fundamental property of methods based on a variational EIG lower bound.
>
> We will clarify this in the paper by revising the methods section to explicitly state the likelihood-free nature of our approach, which allows ALINE to be applied to any simulation-based model [3].
>
> > *3. The paper lacks formal theoretical analysis regarding convergence properties, optimality guarantees, or theoretical relationships to information-theoretic objectives.*
>
> We appreciate your emphasis on theoretical analysis. While our paper does include theoretical contributions, its focus is on the framework and its practical evaluation. In terms of the three points raised:
>
> * **Theoretical Relationships to Information-Theoretic Objectives:** Our framework is built directly upon and formally linked to these objectives. For parameter inference, our acquisition objective (Eq. 8\) is a variational lower bound on the total Expected Information Gain (sEIG). This is a principled approach established in prior work [2, 4]. For predictive tasks, a key contribution of our work is the introduction of the sequential Expected Predictive Information Gain (sEPIG). In **Proposition 1 (with proof in Appendix A.2)**, we formally prove that our proposed objective (Eq. 9\) is a valid variational lower bound on this new information-theoretic quantity.
> * **Optimality Guarantees:** **Proposition 2** formally shows that the gap between our objective and the true information gain is precisely the KL divergence between our model's approximation and the true posterior. This means that in the limit of a perfect inference network (i.e., when the KL divergence is zero), our policy is optimized to maximize the **true sequential information gain** – which is the optimal criterion from an information-theoretic standpoint. While finding the globally optimal policy for this objective is generally intractable, our RL framework is a standard method for discovering high-performing policies that maximize this principled objective.
> * **Convergence Properties:** The primary contribution of this paper is the unified and flexible framework for joint acquisition and inference, rather than developing a new reinforcement learning algorithm. Therefore, a formal convergence analysis of the specific RL algorithm is outside the scope of our contributions. We employed the standard REINFORCE algorithm for its simplicity and effectiveness. Its convergence properties can be referenced from existing works, such as [5].
>
> We hope this clarifies the extent and focus of our theoretical contributions.
>
> > *4. (typo) Line 210: Should the "locations" here be deleted as not specifically discussing the Location Finding task here?*
>
> Thank you for flagging this potential ambiguity. Our use of the term "locations" here was in the general sense of "input points" {$x^\*\_m$​}, a common term in the literature for points in a function's domain. However, we agree that this could be confusing given the specific "Location Finding" task discussed elsewhere. To prevent any misunderstanding, we will revise the manuscript and replace "locations" with the more general term **"inputs"** in this context.
>
> **References**
>
> [1] Shen et al., Variational sequential optimal experimental design using reinforcement learning, Computer Methods in Applied Mechanics and Engineering 444 (2025).
>
> [2] Blau et al., Statistically Efficient Bayesian Sequential Experiment Design via Reinforcement Learning with Cross-Entropy Estimators, arXiv preprint (2023).
>
> [3] Cranmer et al., The frontier of simulation-based inference, Proceedings of the National Academy of Sciences (2020).
>
> [4] Foster et al., Variational Bayesian Optimal Experimental Design, NeurIPS (2019).
>
> [5] Zhang et al., Sample efficient reinforcement learning with REINFORCE, AAAI (2021).

---

> > ### Comment · Reviewer_gxPy · 2025-08-04
> > **Official Comments by Reviewer gxPy**
> >
> > Thank you for the response and the additional experiments. I increased my score by 1.

---

> > > ### Author Response · Authors · 2025-08-04
> > >
> > > Thank you very much. We greatly appreciate it.

---

### Official Review · Reviewer_9JtZ · 2025-07-08

**Clarity:** 3
**Significance:** 3
**Originality:** 2
**Rating:** 5
**Confidence:** 4

**Summary:**

This paper introduces the **Amortized Active Learning and Inference Engine (ALINE)**, a unified framework for amortized Bayesian inference and active data acquisition that relies on a transformer-based backbone. At run time, ALINE ingests the current data history along with a user-specified target specifier (any subset of parameters or predictive outputs). In a single forward pass it

1. produces an approximate posterior or predictive distribution, and
2. proposes a candidate query point expected to deliver the greatest information gain for the specified target.

Training combines a maximum-likelihood objective for the inference head with a reinforcement-learning objective for the query head, where each step’s reward is the self-estimated improvement in the log-probability of the chosen targets.

Empirically, ALINE matches or exceeds strong baselines—such as Gaussian processes with uncertainty sampling or expected information gain—on standard active-learning benchmarks and outperforms them on out-of-distribution tasks. It also achieves competitive performance on classical Bayesian experimental-design problems (e.g., Location-Finding, CES, and psychometric-model tasks) while remaining orders of magnitude faster at deployment than non-amortized alternatives.

**Questions:**

1. **Equation 6 appears to be incorrect.** It should be expressed in an autoregressive form, because the terms are not conditionally independent given the dataset.

2. **Why are the parameters and their inference not modeled autoregressively?** An autoregressive treatment seems the principled approach.

3. **Consider relocating the Related-Work section to follow the Introduction** to improve the paper’s narrative flow.

**Ethical Concerns:**

["NO or VERY MINOR ethics concerns only"]

**Final Justification:**

Author's detailed rebuttal has addressed most of my questions, so I’m raising my score by 1.

**Limitations:**

Yes, the authors have discussed the limitations of their work.

**Paper Formatting Concerns:**

There are no significant issues with the paper’s formatting.

**Quality:**

3

**Strengths And Weaknesses:**

**STRENGTHS**

1. Provides a unified amortization framework for both inference and acquisition. The target-specifier mechanism lets a single trained model focus on any subset of parameters or predictions at run time.

2. The paper is clearly written and well organized, making it easy to follow.

3. Delivers strong empirical performance while enabling much faster deployment.

**WEAKNESSES**

1. The empirical study is confined to Gaussian-process benchmarks; broader experiments would strengthen the evidence.

2. The rationale for combining inference and active data acquisition is not fully developed. What, exactly, would be lost by handling them separately, and what concrete computational or practical gains come from unifying them?

3. Correct joint inference requires autoregressive prediction, yet this is deferred to future work.

---

> ### Author Rebuttal · Authors · 2025-07-30
>
> Thank you for your positive assessment of our work and the points you raised. We address your points and questions below.
>
> > *1. The empirical study is confined to Gaussian-process benchmarks; broader experiments would strengthen the evidence.*
>
> We would like to clarify that our evaluation already included diverse, non-GP settings, such as two classical BED benchmarks (Location Finding and CES) and a realistic psychometric modeling task with domain-specific baselines. We will mention more clearly in the paper.
>
> Nevertheless, we fully agree that demonstrating scalability on even more complex problems is an excellent suggestion - a point also raised by other reviewers. To that end, we have conducted a new set of experiments on a practical high-dimensional task: **Active Exploration of Hyperparameter Performance Landscapes**. This experiment aims to efficiently characterize a machine learning model's overall behavior on a new task, allowing practitioners to quickly assess a model family's viability or understand its sensitivities. The task is to actively query a small number of hyperparameter configurations to build a surrogate model that accurately predicts performance for a larger, held-out set of target configurations.
>
> We used the high-dimensional, real-world tasks from the **HPO-B meta benchmark** [1]. Specifically, we evaluated on the *rpart* (6D), *svm* (8D), *ranger* (9D), and *xgboost* (16D) datasets. ALINE was trained on their multiple pre-defined training sets. We then evaluated its performance, alongside non-amortized GP-based baselines and an amortized surrogate baseline (ACE-US), on the benchmark's held-out and entirely unseen test sets. This setup is explicitly designed to test the model's ability to generalize to new problems.
>
> The table below shows the RMSE with 95% confidence interval after 30 steps, averaged across all test tasks for each dataset. First, **both amortized methods, ALINE and ACE-US, significantly outperform all non-amortized GP-based baselines** across all tasks. This highlights the power of meta-learning in this domain. GP-based methods must learn each new performance landscape from scratch, which is highly inefficient in high dimensions. In contrast, both ALINE and ACE-US are pre-trained on hundreds of related tasks and their Transformer architectures meta-learn the structural patterns common to these landscapes. This shared prior knowledge allows them to make far more accurate predictions from sparse data. Second, while ACE-US performs strongly due to its amortized nature, **ALINE consistently achieves the best or joint-best performance**. This demonstrates the additional, crucial benefit of our core contribution: the learned acquisition policy. ACE-US relies on a standard heuristic, whereas ALINE's policy is trained end-to-end to learn how to optimally explore the landscape, leading to more informative queries and ultimately a more accurate final surrogate model.
>
> We believe this new experiment, alongside the existing BED and psychometric tasks, now provides a comprehensive and compelling demonstration of ALINE's practical utility. We will add these results to the revised manuscript.
>
> | RMSE ($\\downarrow$) | GP-RS | GP-US | GP-VR | GP-EPIG | ACE-US | ALINE |
> | :---- | :---- | :---- | :---- | :---- | :---- | :---- |
> | rpart (6D) | 0.07±0.03 | 0.04±0.02 | 0.04±0.02 | 0.05±0.02 | **0.01±0.00** | **0.01±0.00** |
> | svm (8D) | 0.22±0.11 | 0.11±0.05 | 0.12±0.07 | 0.15±0.08 | 0.04±0.01 | **0.03±0.01** |
> | ranger (9D) | 0.10±0.02 | 0.07±0.01 | 0.08±0.02 | 0.08±0.02 | **0.02±0.01** | **0.02±0.01** |
> | xgboost (16D) | 0.09±0.02 | 0.09±0.02 | 0.09±0.02 | 0.09±0.02 | 0.04±0.01 | **0.03±0.01** |
>
>
> > *2. The rationale for combining inference and active data acquisition is not fully developed. What, exactly, would be lost by handling them separately, and what concrete computational or practical gains come from unifying them?*
>
> We appreciate the opportunity to elaborate on the concrete benefits of combining inference and active data acquisition. Our rationale is grounded in two core advantages:
>
> * **Significant Gains in Computational and Parameter Efficiency**: The most direct advantage of our unified framework is its efficiency. Both the inference task and the acquisition policy rely on the exact *same context history*. Our unified framework uses a *single, shared Transformer backbone* to process this context data. In contrast, a separate approach would lead to a near-doubling of model parameters, significantly increasing training costs and deployment latency, as two separate forward passes would be required for each step.
> * **Improved Representation via Joint Learning**: Secondly, we believe our joint training framework could benefit from principles of multi-task learning (MTL) \[2, 3\]. The rationale is that the two tasks provide complementary learning signals that can mutually regularize the shared backbone. The inference objective pushes the model to learn underlying data patterns for accurate predictions, while the acquisition objective forces it to learn a robust representation of its own uncertainty to make informative queries. Forcing the model to learn a single representation that serves both objectives is a form of regularization that encourages a more generalizable understanding of the problem, compared to training on either task alone.
>
> We will add the above discussion to both the method and the discussion section in the revised manuscript.
>
>
> > *3. Correct joint inference requires autoregressive prediction, yet this is deferred to future work. Why are the parameters and their inference not modeled autoregressively? An autoregressive treatment seems the principled approach.*
>
> Good point. Our decision to adopt a factorized approximation was a deliberate and critical **trade-off between modeling accuracy and computational tractability**. First, an autoregressive model would require sequential generation of each marginal dimension of the posterior, with each step conditioned on the previously generated ones. To compute the joint posterior's log-probability for $M$ targets, this would necessitate **$M$** **sequential forward passes** through the Transformer network. Since the reward for our policy agent is the improvement in this log-probability, calculating this reward at every single step of every training episode would become prohibitively expensive, as the cost would scale linearly with the number of targets. This would make training the acquisition policy **infeasibly slow**, especially for problems with many parameters or prediction targets. Therefore, ALINE adopts the mean-field approximation, which is a common and established practice in the Neural Process literature \[4, 5, 6\]. This allows for efficient, parallel computation of the marginals in a single forward pass. Also, our experiments demonstrate that even with this factorized assumption, ALINE achieves strong empirical performance, often outperforming competitive baselines. This suggests that for the tasks evaluated, the benefits gained from our joint amortization and flexible targeting strategy outweigh the limitations of the structural approximation of the posterior.
>
> We agree that an autoregressive approach represents the most principled way to model the joint posterior distribution. We share your belief that developing more efficient methods for autoregressive inference in such models is a crucial and exciting direction for future research. As you rightly suggest, such advances are likely essential for scaling up to the next generation of foundation models in this field. For this work, however, we focused on establishing the core viability of the unified ALINE framework itself. We will add a discussion of this trade-off to the revised manuscript to make our design rationale clearer.
>
> > *4. Equation 6 appears to be incorrect. It should be expressed in an autoregressive form, because the terms are not conditionally independent given the dataset.*
>
> Thank you for raising this point. It is true that, in the general case, the posterior predictive outputs​ are not conditionally independent given the history, and thus the true objective should be based on the joint distribution. Our current Equation 6 is derived under the explicit assumption of a **factorized likelihood**, a choice we mention on line 144. This is a common and practical simplification in the Neural Process literature \[4, 5, 6\], adopted to maintain computational tractability.
>
> But yes, the presentation could be clearer. We will revise to stating the objective first in a form that does not assume factorization, and then explicitly state that for practical implementation, we approximate this joint distribution with a factorized model.
>
> > *5. Consider relocating the Related-Work section to follow the Introduction to improve the paper’s narrative flow.*
>
> Good idea; we will update accordingly.
>
> **References**
>
> [1] Arango, et al., HPO-B: A Large-Scale Reproducible Benchmark for Black-Box HPO based on OpenML, NeurIPS Track on Datasets and Benchmarks (2021).
>
> [2] Liu, et al., Multi-task deep neural networks for natural language understanding, ACL (2019).
>
> [3] Yu, et al., Unleashing the Power of Multi-Task Learning: A Comprehensive Survey Spanning Traditional, Deep, and Pretrained Foundation Model Eras, arXiv preprint (2024).
>
> [4] Garnelo, et al., Conditional neural processes, ICML (2018).
>
> [5] Müller, et al., Transformers Can Do Bayesian Inference, ICLR (2022).
>
> [6] Chang, et al., Amortized Probabilistic Conditioning for Optimization, Simulation and Inference, AISTATS (2025).

---

> > ### Comment · Reviewer_9JtZ · 2025-08-05
> >
> > Thank you for the detailed response—it addressed most of my questions, so I’m raising my score by 1.

---

> > > ### Author Response · Authors · 2025-08-05
> > >
> > > Thank you for your reply and for raising the score. We really appreciate it.

---

### Note · Authors · 2025-08-11

We thank all reviewers for their time and valuable feedback. We are delighted that, following a productive discussion period, all reviewers have provided a positive evaluation of our work. Here, we briefly summarize the key improvements made in response to the suggestions.

### New Experimental Results

* Added a new set of high-dimensional (up to 16D) "**Active Learning for Hyperparameter Exploration**" experiments on the HPO-B benchmark. (Reviewers 9JtZ, xxdt, GfQV).
* Added **vsOED** as a new baseline for BED experiments. (Reviewer gxPy).
* Conducted two new investigations on the psychometric model to demonstrate ALINE's key features: (1) dynamically **switching targets** mid-rollout and (2) **generalizing to novel target combinations** not seen during training (Reviewers xxdt, GfQV).

### Clarifications and Manuscript Improvements

* Added discussion on the trade-offs of **autoregressive modeling**, clarifying our design choice for computational tractability (Reviewers 9JtZ, xxdt).
* Expanded the **related work** section to include and discuss the valuable literature suggested by the reviewers (Reviewers gxPy, GfQV).
* All other clarifications promised during the rebuttal (e.g., on implicit likelihood training, the sim2real gap, and the gradient flow during joint training) will be incorporated into the final manuscript.

We are very grateful to the reviewers for their thorough and constructive engagement. Their feedback has been invaluable and has helped us significantly improve the clarity and empirical validation of our work.

---

### Decision · Program_Chairs · 2025-09-17

**Decision:**

Accept (spotlight)

**Comment:**

The paper proposes ALINE, a unified framework for amortized Bayesian inference and active data acquisition built on a transformer backbone with joint inference–acquisition training. The method introduces a flexible target-specification mechanism and demonstrates strong empirical performance across regression-based active learning, Bayesian experimental design benchmarks, and psychometric tasks. Reviewers appreciated the clarity, the efficiency of the unified architecture, and the novelty of the flexible targeting strategy. The main concerns before the rebuttal phase centered on overlap with prior work (vsOED), reliance on a factorized likelihood approximation, and the sim2real training assumption. The authors addressed these with additional experiments on high-dimensional benchmarks, direct comparisons to vsOED, and clarifications of design trade-offs and limitations.

Overall, I find the paper technically solid, with consensus among reviewers leaning toward acceptance. The authors are encouraged to strengthen the discussion of limitations and clarify distinctions from prior work in the final version.